# Hydrodynamics without Averaging – a Hard Rods Study

**Friedrich Hübner**[1*]

**1** Department of Mathematics, King's College London, Strand WC2R 2LS, London, U.K.

* friedrich.huebner@kcl.ac.uk

## Abstract

On the example of the integrable hard rods model we study the quality of the (generalized) hydrodynamic approximation on a single coarse-grained sample. This is opposed to the traditional approach which averages over an appropriate local equilibrium state. While mathematically more ambiguous, a major advantage of the new approach is that it allows us to disentangle intrinsic diffusion from 'diffusion from convection' effects. For the hard rods we find intrinsic diffusion is absent, which agrees with and clarifies recent findings. Interestingly, the results also apply to not locally thermal states, demonstrating that hydrodynamics (in this model) does not require the assumption of local equilibrium.

# 1   Introduction

Effective descriptions are important tools to simplify and subsequently study otherwise intractable many-body systems. The hydrodynamic approximation [1,2] is particularly important as it describes finite density system on large spatial and time scales, a regime often well justified in real life systems. Hydrodynamics allows us to describe the state of a system by only looking at the distribution $q_n(x)$ of its conserved quantities (here $n$ labels the conserved quantity)[1], a drastic simplification compared to microscopic dynamics. The evolution equations are then simply given by the continuity equations

$$\partial_t q_n(t,x) + \partial_x j_n(t,x) = 0. \tag{1}$$

Note that crucially these are not the microscopic continuity equations. Equation (1) holds microscopically, however, the problem is that the microscopic currents $j_n(t,x)$ depend on the precise microscopic state and thus cannot be computed from knowing $q_n(t,x)$ alone. One overcomes this problem phenomenologically by assuming that the state is a slowly evolving local equilibrium state

$$f \sim e^{-\int dx \sum_n \beta_n(t/\ell, x/\ell) q_n(t,x)}, \tag{2}$$

described by local Lagrange multipliers $\beta_n(t,x)$. Here $\ell \gg 1$ is a large length scale, ensuring that the system is locally in equilibrium, i.e. $\beta_n(t,x) \approx$ const (the local state is also called a generalized Gibbs ensemble – GGE). Therefore, given the local charge densities $q_n(t,x)$, we can compute their currents $j_n(t,x)$ as expectation values in the GGE state whose $\beta_n(t,x)$ correspond to $q_n(t,x)$. This way the equations become closed set of PDE's. When applied to fluids, this reasoning gives rise to the Euler equation of fluid dynamics, hence the name hydrodynamics.

The Euler equation is generally well accepted (and also verified both experimentally and numerically) to be a good description in the Euler scaling limit. In this scaling limit the length and time scales of observation as well as the particle number $\ell \sim T \sim \to \infty$ are send to infinity proportionally, meaning hydrodynamics describes finite density states and finite velocity dynamics. Nonetheless, our theoretical understanding remains unsatisfactory. In particular,

---

[1]In a typical Galilei invariant system there are three such conserved quantities: particle number, momentum and energy.

why are we able to replace the true state of the system by local GGE states? The fundamental problem is that thermalization does not truly drive the state of the system towards a local GGE state, simply because this would increase entropy: in a classical system a single configuration of particles has to remain a single configuration of particles and in a quantum system a pure quantum state has to remain pure. It is by now understood that local observables indeed appear to be close to their GGE expectation values after sufficiently long times, but global observables do not (they still retain all information about the initial state) [3–12]. Luckily, both $q_n(t, x)$ and $j_n(t, x)$ are local observables, hence (1) is well justified in the (Euler) hydrodynamic scaling limit.

However, hydrodynamic equations like (1) generically tend to break down after a finite time by developing gradient catastrophes (shocks in 1D [13] and turbulence in higher D [14]). The generic wisdom is that these should be healed by incorporating the diffusive correction $j_n \to j_n + \sum_m D_{nm} \partial_x q_m$ into hydrodynamics (in usual Galilei invariant systems this is the Navier-Stokes equation) [1, 2, 13]. As it is proportional to $\partial_x q_n$ it is negligible in the large scale limit. However, it becomes relevant close to the development of gradient catastrophes and smoothens them out. Naturally, the diffusion matrix can be computed from the local GGE correlation functions using the Kubo-formula. At this point, however, the validity of the local equilibrium assumption is unclear: we cannot necessarily infer it from the validity of the Euler equation, since correlations are suppressed in the Euler scaling limit (i.e. even if the correlations would be differ from GGE correlations, the Euler equations would still emerge). Hence, there is no reason why the true correlations should be described by the local GGE correlations. Note that even if one would insist that correlations should be thermodynamic it would not be clear which thermodynamic ensemble to use: for instance, the microcanonical and canonical ensembles have different correlations.

Furthermore, hydrodynamics can also be applied to non-standard settings: for instance, it has been proposed to be able to predict the evolution of (large scale averaged) correlation functions using Ballistic Macroscopic Fluctuation Theory – BMFT [15, 16]. This predicts that long range correlations will emerge in the system over time due to ballistic transport of initial fluctuations, meaning that even if one starts with an (uncorrelated) state like (2), the system cannot be in such a state at later times. It has been proposed under the name "diffusion from convection" [17, 18] that such correlations give rise to additional terms on the diffusive scale as well (but these terms are not necessarily of the same form as terms due to intrinsic diffusion).

Therefore, it is not really clear that the artificially introduced randomness by considering a state like (2) as initial state and by adhoc replacing the evolved local state by a local GGE state Navier-Stokes indeed gives the correct diffusive correction. The same holds for adding explicit randomness into the dynamics of the system (to model effective noise in the system), which mathematically simpler to study as it explicitly drives the system towards local equilibrium (a well known paradigmatic many body system with noise is for instance TASEP [19, 20]).

To get rid of all of these artificial effects, we propose a radically new approach to understand hydrodynamics called "Hydrodynamics without averaging". The idea is to understand the emergence of hydrodynamics fully deterministically: the initial state is a single deterministic configuration. From this initial state we obtain an initial state for the hydrodynamic equation via coarse-graining. Then the prediction of the hydrodynamic equation is compared to the deterministic microscopic dynamics of the system. The reason why (1) emerges is due to self-averaging of the microscopic details. Conceptually, most of the ideas on which "Hydrodynamics without averaging" is based are well known in the literature and have been applied as guiding principles (for instance, it is the work horse behind BMFT). What is radical about this concept/work is that it has not been carried out in an actual model. It is also radical in the sense that it is mathematically vague[2]. Nevertheless, we will obtain important insights:

---

[2]One important reason why one considers an ensemble of initial configurations is that one can (in principle)

one of the main insights we would like to put forward is that already the unavoidable initial time coarse-graining introduces an error that will provide a hard bound for the precision of the hydrodynamic approximation. This error will depend on the coarse-graining scheme, but seems to be always bigger than $\mathcal{O}(1/\ell^2)$. Hence, in principle, hydrodynamic theories should be able to capture diffusive corrections, but likely no higher order effects.

Since we are far from being able to carry out such a procedure for an actual fluid, we develop this approach on the example of the exactly solvable one-dimensional hard spheres model [20, 21], also called hard rods model. A main motivation for this work was to understand the leading order correction term to hydrodynamics: the diffusive $1/\ell$ correction. Recently it was discovered [18, 22] that this is not given by the usual entropy increasing diffusive term [23], but instead by complicated and unconventional entropy conserving terms. Jugding from the form of the equations they seem to be explained by "diffusion from convection", but a precise understanding is hindered by the fact that all statements are always averaged over the initial state. Therefore, "hydrodynamics without averaging" is ideal to study this problem as all "diffusion from convection" contributions are absent. Only a true intrinsic diffusion can affect the equations. In this work we find that there is no such intrinsic diffusion: Euler hydrodynamics is valid even at the diffusive scale. This is both surprising (physically there should be diffusion) as well as in perfect agreement with the newly found unconventional correction [18, 22]: since there is no intrinsic diffusion all effects on the diffusive scale are due to transport of the initial state randomness, i.e. "diffusion from convection". Such initial state randomness can straight-forwardly taken into account in the "hydrodynamics without averaging" picture. We perform this computation in section 7 and recover the equation found in [18, 22].

At this point we should point out that the hard rods model is an integrable model with an infinite amount of conserved quantities. Therefore, the hydrodynamic equations are far from the usual hydrodynamic equations of a Galilean fluid. They have first been derived in [24, 25]. Later it was found that the hydrodynamic equation of integrable models in general are very similar to those of hard rods. The framework of hydrodynamics in integrable models is nowadays called "Generalized hydrodynamics" (GHD) [26–28] (also see reviews [2, 21, 29–33]). Despite (or even due to) the unusual hydrodynamic equations, GHD has been crucial in the recent years to gain deeper understanding into the principles behind the emergence of hydrodynamics (for instance for the development of "diffusion from convection" or BMFT). GHD is also special as it is linearly degenerate: it does not form shocks like generic 1D hydrodynamics [2, 34, 35, 35, 36] and the next order correction scales indeed diffusively (unlike generic 1D models which have KPZ-like fluctuations [37]). In both fields, integrability and hydrodynamicss, the hard rods model has always been fundamental, due to its simplicity and due to the fact that its dynamics can be solved exactly. Therefore, it is still an actively studied model, see for instance the recent developments in [23, 38–49]. In particular, its hydrodynamics (in the sense of averaging over an initial state) is rigorously proven [25] (this is also the first proof of emergence of hydrodynamics in any interacting Hamiltonian model). Therefore, this model is an ideal starting point to understand the features of "hydrodynamics without averaging" and compare to the usual "hydrodynamics with averaging".

The paper is organized as follows. In section 2 we review the hard rods model and its hydrodynamics. In section 3 we then carry out "hydrodynamics without averaging" using fluid cell coarse-graining and derive an analytic prediction of the scaling of the error of the hydrodynamic approximation. We repeat the same analysis on a different coarse-graining scheme (smoothing) in section 4. In section 5 we then explain how the usual diffusive scale correction to GHD can be obtained by averaging the GHD equation over a random initial state. Note that the Euler hydrodynamic equations are time-reversible (due to the absence of shocks) [2], in

---

clearly quantify the precision of the hydrodynamics approximation as function of $\ell$.

particular they conserve entropy. It was believed that diffusion would lead to entropy increase and eventually thermalization (to a GGE). Since this paper shows that such intrinsic diffusion is absent, we reinvestigate the question of entropy increase and thermalization in section 6. We propose that it is due to coarse-graining. In particular the thermalization time scale is observer dependent and should occur before the diffusive time scale.

## 2    The hard rods model

Hard rods are hard spheres in one dimension [20], with Hamiltonian given by

$$H(\vec{x}, \vec{p}) = \tfrac{1}{2} \sum_i p_i^2 + \sum_{i \neq j} V(x_i - x_j), \tag{3}$$

where $V(x) = \infty$ if $|x| < a$ and otherwise 0. Here $a$ is the hard rod diameter. Particle trajectories are simple: they evolve like free particles $\frac{\mathrm{d}}{\mathrm{d}t} x_i = p_i$ until they hit another particle. During scattering both particles instantaneously exchange their momenta $p_i \leftrightarrow p_j$ (like billiard balls), see Fig. 1 a). Thus, the number of particles with each momentum $p$ is conserved.

Instead of a discrete index $n$ we thus have a continuum $p \in \mathbb{R}$ index labeling the conserved quantities. Instead of $q_n(t, x)$ one uses the quasi-particle density $\rho(t, x, p)$, which is interpreted as the density of quasi-particles with momentum $p$ at position $x$ at time $t$. These quasi-particles are not the physical particles, which swap momenta during scattering. Instead the quasi-particles (or tracer particles), keep their momenta and swap their positions, i.e. both particle jump forward by $a$ (this is merely a relabeling of particles during each scattering), see Fig. 1 b). We will denote by $x_i(t)$ the location of the $i$'th tracer particle.

In order to obtain the hydrodynamics of this model we need to study a large scale initial configurations. By this we mean that the particles are distributed over a large length scale $\ell \gg a$ with a finite density, i.e. a large number of particles $N \sim \ell$. We will also study the system at times proportional to $\ell$. To keep consistent notation throughout the paper, we will use macroscopic variables from now on, i.e. quantify space and time in units of $\ell$. This is equivalent to replacing $a \to a/\ell$.

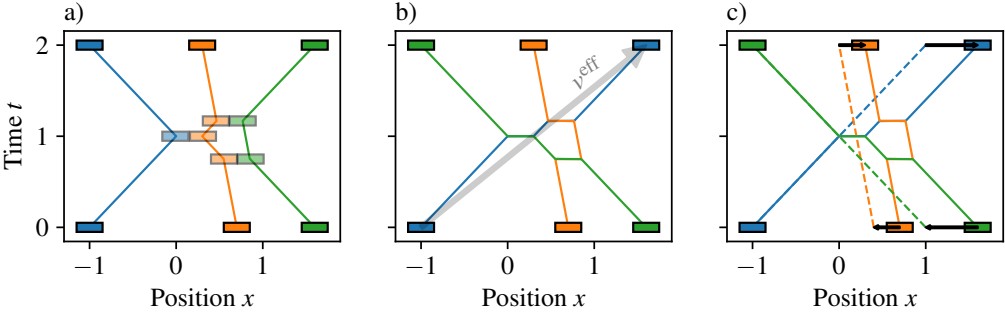

Figure 1: Dynamics of hard rods: a) physical hard rods scatter by exchanging their momenta $p$. b) Alternatively, particles exchange positions instead of momenta during scattering. Therefore, particles effectively travel a larger distance in a given time interval, which is observed as an effective velocity in GHD. c) Using the contraction map (4) hard rods can be mapped to non-interacting particles. In these coordinates evolution is trivial. To obtain the location of hard rods at a later time one simply has to expand back to original coordinates.

The advantage of using tracer particles is that their dynamics can be explicitly computed:

$$\hat{x}_i = x_i - \frac{a}{\ell}\sum_{j\neq i}\theta(x_i - x_j) \tag{4}$$

$$\hat{x}_i(t) = \hat{x}_i + p_i t \tag{5}$$

$$x_i(t) = \hat{x}_i(t) + \frac{a}{\ell}\sum_{j\neq i}\theta(\hat{x}_i(t) - \hat{x}_j(t)). \tag{6}$$

This solution is based on the "space contraction" (4), which maps the system onto non-interacting particles [25], see Fig. 1 c). This space contraction removes the size of the hard rods, i.e. the first particle on the left is untouched, the second particle is moved by $-a$, the third by $-2a$ and so on. In contracted coordinates $\hat{x}_i$ the evolution is trivial. Thus one only has to re-expand this configuration at final time. The existence of such a simple explicit solution is what makes hard rods so unique among interacting many-body systems.

In the hydrodynamic limit $\ell \to \infty$, we define the quasi-particle density as

$$\rho(t,x,p) = \frac{1}{\ell}\sum_i \delta(x - x_i(t))\delta(p - p_i). \tag{7}$$

Its Euler scale hydrodynamic equation is given by [25]

$$\partial_t \rho(t,x,p) + \partial_x(v^{\text{eff}}(t,x,p)\rho(t,x,p)) = 0, \tag{8}$$

where the effective velocity

$$v^{\text{eff}}(t,x,p) = \frac{p - a\int \mathrm{d}q\, q\rho(t,x,q)}{1 - a\int \mathrm{d}q\, \rho(t,x,q)} \tag{9}$$

is interpreted as the effective velocity of the quasi-particle incorporating the jumps by scattering to other particles, see Fig. 1 b). Note that by integrating (8) over $x$ we find that $Q(p) = \int \mathrm{d}x\, \rho(t,x,p)$ is conserved in time. These are the infinitely many conservation laws present in this model.

Equation (8) has an explicit solution starting from initial density $\rho(x,p)$ at $t = 0$: the trajectory of the quasi-particle $X(t,x,p)$ starting at $x$ with momentum $p$ is given by $X(t,x,p)$

$$X(t,x,p) = \hat{X}(x) + a\int \mathrm{d}y\,\mathrm{d}q\, \rho(y,q)\theta(\hat{X}(x) + pt - \hat{X}(y) - qt) - \frac{a}{2\ell}, \tag{10}$$

$$\hat{X}(x) = x - a\int \mathrm{d}y\int \mathrm{d}q\, \rho(y,q)\theta(x - y) + \frac{a}{2\ell}, \tag{11}$$

from which the solution can be obtained by the push-forward $\rho(t,\cdot,p) = X(t,\cdot,p)_*\rho(\cdot,p)$, i.e. for any observable $\Upsilon(x,p)$ we have

$$\langle \rho(t), \Upsilon \rangle := \int \mathrm{d}x\,\mathrm{d}p\, \rho(t,x,p)\Upsilon(x,p) = \int \mathrm{d}x\,\mathrm{d}p\, \rho(x,p)\Upsilon(X(t,x,p),p), \tag{12}$$

or more explicitly $\rho(t,X(t,x,p),p) = \frac{\rho(x,p)}{\partial_x X(t,x,p)}$ from which $\rho(t,x,p)$ can be obtained by inverting the monotone increasing function $X(t,x,p)$ in $x$. Equations (11-10) are simply the continuum versions of (4-6). Note that the additional constant $\frac{a}{2\ell}$ in (11) was chosen for later convenience.

The fact that in hard rods we both have an explicit solution to their microscopic dynamics and to their hydrodynamics made the hards model one of the most important models to study the emergence of hydrodynamics.

## 3  Coarse-graining 1: Fluid cell averaging

A crucial observation is that the hydrodynamic equation (8) does not make any sense when applied to a microscopic $\rho(t,x,p)$ as in (7), simply because it is spiky. To overcome this problem, one usually averages the initial $\rho(x,p)$ over some ensemble of states, for instance given by (2), and obtains

$$\langle \rho(x,p) \rangle = \left\langle \frac{1}{\ell} \sum_i \delta(x-x_i)\delta(p-p_i) \right\rangle, \tag{13}$$

which is now a smooth function. Then (8), with the replacement $\rho(t,x,p) \to \langle \rho(t,x,p) \rangle$, makes sense. In this regard (8) was rigorously proven for a large family of states in [25]. However, as said in the introduction, this works on the Euler scale, but the randomness of the initial state might affect higher order corrections. Therefore, we would like to avoid this procedure.

Instead, the procedure proposed here is to take a fixed initial configuration. In order to obtain any meaningful initial density $\rho(x,p)$ for the hydrodynamic theory one has to coarse-grain (7) over a mesoscopic scale $\Delta x$ and $\Delta p$. While this is, at least in spirit, generally thought to be the correct procedure to obtain hydrodynamics, it actually poses non-trivial conceptual problems

- The precision of hydrodynamics depends on the arbitrarily chosen coarse-graining scales $\Delta x$ and $\Delta p$. In an experiment it is natural to consider them to be given by the precision of the measurement device. Nevertheless, this procedure makes hydrodynamics observer dependent.

- The initial density $\rho(x,p)$ after coarse-graining will be rough on the macroscopic scale[3]. In particular, applying (8) does not make sense as $\rho(t,x,p)$ will not be differentiable. Luckily, the explicit solution (10) also makes sense for rough initial data. However, the solution will in principle be only a weak solution to the hydrodynamic equation. In integrable systems, such as hard rods, weak solutions are unique[4] [22, 34]. In non-integrable models they do not have to be after developing gradient catastrophes, but schemes exists to identify the physical solution.

From a practical perspective it is not a trivial task to quantify the quality of the hydrodynamic approximation. We will do this by comparing the value of a (smooth) observable $\Upsilon(x,p)$ in the microscopic theory to the one obtained by the hydrodynamic prediction. Intuitively, as $\ell \to \infty$ the difference should decay to 0. However, as we change $\ell$, we also need to adjust the number of particles accordingly. Therefore, for each $\ell$ we have a completely different configuration, that somehow in the limit should approach a smooth distribution. Additionally, it is clear that not all configurations are described by hydrodynamics. It will be possible to come up with some fine-tuned configurations that violate hydrodynamics[5]. Hydrodynamics will only work if locally the distribution of particles is sufficiently generic. Mathematically speaking one would say that hydrodynamics will work for configurations almost surely with respect to a measure like (2). While this idea is useful, also for numerical checks, it almost surely can only be defined w.r.t. a probability measure: we want to achieve an even more

---

[3]Of course, one can choose a smooth coarse-graining on scale $\Delta x$. However, as $\Delta x \to 0$, this will still have large gradients $\partial_x \rho(x,p) \sim 1/\Delta x$ on the macroscopic scale.

[4]To be more precise: this is known in hard rods and the Lieb-Liniger model, but expected to be true more generally.

[5]We are not aware of an explicit construction of such hydrodynamics violating states, but this is generally expected.

general statement, namely that hydrodynamics works for almost surely w.r.t any physical and sufficiently generic measures.

Due to the many conceptual and practical problems, the derivations in this paper will be far from rigorous. Nevertheless, we will be able to quantify the precision of the hydrodynamic approximation in a meaningful manner (which will also agree with numerical simulations). For future applications of similar ideas it would be fundamental to develop a better mathematical framework in which these computations could be done more reliably. We leave this as an open problem.

After this heuristic overview, we now move on to carrying out the procedure in hard rods.

## 3.1 Initial state coarse-graining

We will start by looking at a very basic, but already quite insightful problem: In order to compare the microscopic dynamics with the hydrodynamic evolution $\rho(t, x, p)$, we first need to define what initial $\rho(x, p)$ we associate to an initial configuration $x_i, p_i$.

A trivial guess would be to use the empirical density of (7), i.e.

$$\rho_{\text{Micro}}(x, p) = \tfrac{1}{\ell} \sum_i \delta(x - x_i)\delta(p - p_i).\tag{14}$$

As mentioned above, the problem is that is that for finite $N$ this will always be a 'spiky' function. Instead, we need to find a continuous approximation to (14). In this paper we will investigate two ways to do that.

The first corase-graining scheme is fluid cell averaging. This is a natural concept in hydrodynamics: let us divide the $x, p$ plane into boxes (aka fluid cells) of size $\Delta x \times \Delta p$. To be precise, we denote for integers $\alpha$ and $\beta$:

$$A_\alpha = [x_\alpha - \Delta x/2, x_\alpha + \Delta x/2]\tag{15}$$
$$B_\beta = [p_\beta - \Delta p/2, p_\beta + \Delta p/2]\tag{16}$$
$$C_{\alpha,\beta} = A_\alpha \times B_\beta,\tag{17}$$

where $x_\alpha = \alpha \Delta x$ and $p_\beta = \beta \Delta p$ are the centers of the boxes. Now, let us denote by $n_{\alpha,\beta}$ the number of particles in box $C_{\alpha,\beta}$ (which scales like $\ell \Delta x \Delta p$) and by

$$\rho_{\alpha,\beta} = \frac{n_{\alpha,\beta}}{\ell \Delta x \Delta p} = \frac{1}{\ell \Delta x \Delta p} \sum_i \theta(x_i \in A_\alpha)\theta(p_i \in B_\beta)\tag{18}$$

the density of particles in this region. For abuse of notation we will write $i \in A_\alpha$ iff $x_i \in A_\alpha$, $i \in B_\beta$ iff $p_i \in B_\beta$ and $i \in C_{\alpha,\beta}$ iff $(x_i, p_i) \in C_{\alpha,\beta}$.

The coarse-graining approximation is then given by a constant density $\rho_{\alpha,\beta}$ in each box:

$$\rho_{\text{FC}}(x, p) = \sum_{\alpha,\beta} \theta((x, p) \in C_{\alpha,\beta})\rho_{\alpha,\beta}.\tag{19}$$

We want to make these fluid cells sufficiently big, such that there are many particles in them, but on the other hand they should also be much smaller than the macroscopic scale. Therefore it is natural to assume $1/\ell \ll \Delta x, \Delta p \ll 1$, for instance $\Delta x, \Delta p \sim \ell^{\mu-1}$, where $0 < \mu < 1$ (larger $\mu$ means larger fluid cells). For convenience, to simplify the following discussion, we will assume that $\Delta x \approx \Delta p$.

Let us now try to understand the quality of this approximation. For that let us fix an observable $\Upsilon(x,p)$ and compare the expectation values:

$$\Upsilon_{\text{Micro}} = \frac{1}{\ell} \sum_i \Upsilon(x_i, p_i) \tag{20}$$

$$\Upsilon_{\text{FC}} = \sum_{\alpha,\beta} \rho_{\alpha,\beta} \int_{x_\alpha - \Delta x/2}^{x_\alpha + \Delta x/2} \mathrm{d}x \int_{p_\beta - \Delta p/2}^{p_\beta + \Delta p/2} \mathrm{d}p \, \Upsilon(x,p) \tag{21}$$

$$= \sum_{\alpha,\beta} \rho_{\alpha,\beta} \Delta x \Delta p \int_{-1/2}^{1/2} \mathrm{d}y \, \mathrm{d}q \, \Upsilon(x_\alpha + y\Delta x, p_\beta + q\Delta p). \tag{22}$$

We can write the positions of all the particles inside a fluid cell $\mathsf{C}_{\alpha,\beta}$ as:

$$x_i = x_\alpha + y_i \Delta x \qquad\qquad p_i = p_\beta + q_i \Delta p, \tag{23}$$

where $y_i$ and $q_i$ are between $-1/2$ and $1/2$.

$$\Upsilon_{\text{Micro}} = \frac{1}{\ell} \sum_{\alpha,\beta} \sum_{i \in \mathsf{C}_{\alpha,\beta}} \Upsilon(x_\alpha + y_i \Delta x, p_\beta + q_i \Delta p) \tag{24}$$

$$= \frac{1}{\ell} \sum_{\alpha,\beta} \sum_{i \in \mathsf{C}_{\alpha,\beta}} \Big[ \Upsilon(x_\alpha, p_\beta) + \partial_x \Upsilon(x_\alpha, p_\beta) y_i \Delta x + \partial_p \Upsilon(x_\alpha, p_\beta) q_i \Delta p$$

$$+ \tfrac{1}{2} \partial_x^2 \Upsilon(x_\alpha, p_\beta) y_i^2 \Delta x^2 + \partial_x \partial_p \Upsilon(x_\alpha, p_\beta) y_i p_i \Delta x \Delta p + \tfrac{1}{2} \partial_p^2 \Upsilon(x_\alpha, p_\beta) q_i^2 \Delta p^2 + \mathcal{O}(\Delta x^3) \Big] \tag{25}$$

$$= \frac{1}{\ell} \sum_{\alpha,\beta} n_{\alpha,\beta} \Big[ \Upsilon(x_\alpha, p_\beta) + \partial_x \Upsilon(x_\alpha, p_\beta)[y]_{\alpha,\beta} \Delta x + \partial_p \Upsilon(x_\alpha, p_\beta)[1]_{\alpha,\beta} \Delta p$$

$$+ \tfrac{1}{2} \partial_x^2 \Upsilon(x_\alpha, p_\beta)[y^2]_{\alpha,\beta} + \partial_x \partial_p \Upsilon(x_\alpha, p_\beta)[yq]_{\alpha,\beta} \Delta x \Delta p$$

$$+ \tfrac{1}{2} \partial_p^2 \Upsilon(x_\alpha, p_\beta)[q^2]_{\alpha,\beta} \Delta p^2 + \mathcal{O}(\Delta x^3) \Big]. \tag{26}$$

Here, we introduced the notation to represent averages inside a fluid-cell:

$$[f]_{\alpha,\beta} = \frac{1}{n_{\alpha,\beta}} \sum_{i \in \mathsf{C}_{\alpha,\beta}} f_i. \tag{27}$$

On the other hand we can also derive using Taylor expansion:

$$\Upsilon_{\text{FC}} = \sum_{\alpha,\beta} \rho_{\alpha,\beta} \Delta x \Delta p \Big[ \Upsilon(x_\alpha, p_\beta) + \tfrac{1}{24} \partial_x^2 \Upsilon(x_\alpha, p_\beta) \Delta x^2 + \tfrac{1}{24} \partial_p^2 \Upsilon(x_\alpha, p_\beta) \Delta p^2 + \mathcal{O}(\Delta x^4) \Big] \tag{28}$$

$$= \Delta x \Delta p \sum_{\alpha,\beta} \rho_{\alpha,\beta} \Upsilon(x_\alpha, p_\beta) + \mathcal{O}(\Delta x^2). \tag{29}$$

Therefore, the difference is:

$$\Upsilon_{\text{FC}} - \Upsilon_{\text{Micro}} = \Delta x \Delta p \sum_{\alpha,\beta} \rho_{\alpha,\beta} \Big[ -\partial_x \Upsilon(x_\alpha, p_\beta)[y]_{\alpha,\beta} \Delta x - \partial_p \Upsilon(x_\alpha, p_\beta)[q]_{\alpha,\beta} \Delta p$$

$$+ \tfrac{1}{2} (\tfrac{1}{12} - [y^2]_{\alpha,\beta}) \partial_x^2 \Upsilon(x_\alpha, p_\beta) \Delta x^2 + \tfrac{1}{2} (\tfrac{1}{12} - [q^2]_{\alpha,\beta}) \partial_p^2 \Upsilon(x_\alpha, p_\beta) \Delta p^2$$

$$- \partial_x \partial_p \Upsilon(x_\alpha, p_\beta)[yq]_{\alpha,\beta} \Delta x \Delta p + \mathcal{O}(\Delta x^3) \Big]. \tag{30}$$

Note that, since $\big|[y]_{\alpha,\beta}\big| < 1/2$ and $\big|[q]_{\alpha,\beta}\big| < 1/2$ in case $|\partial_x \Upsilon| < C$ and $\big|\partial_p \Upsilon\big| < C$ is bounded the first order term is strictly bounded by $\frac{1}{2} C(\Delta x + \Delta p) N/\ell$, hence we find:

$$\Upsilon_{\text{FC}} - \Upsilon_{\text{Micro}} = \mathcal{O}(\Delta x). \tag{31}$$

But this is only the worst case scenario, where $[y]_{\alpha,\beta}$ and $[q]_{\alpha,\beta}$ are of $\mathcal{O}(1)$. While it is of course possible to arrange some configurations like that, in a sufficiently generic configurations both $[y]_{\alpha,\beta} \approx [q]_{\alpha,\beta} \approx 0$ should average to almost 0 in each cell. Even if there is one cell, where $[y]_{\alpha,\beta}$ is a $\mathcal{O}(1)$ positive number, then there will be other cell, where it is a negative $\mathcal{O}(1)$ positive number. Hence, it is natural to assume that generically the difference will be much lower.

At this point we need to employ intuition about the self-averaging of terms like $[y]_{\alpha,\beta}$. They are a sum of random numbers $y_i = \mathcal{O}(1)$ of no clear sign. If there is no bias in the $y_i$, from the central limit theorem we can estimate that the size of this term is $\sum_{i \in C_{\alpha,\beta}} y_i \sim \pm\sqrt{n_{\alpha,\beta}}$ $\sim \pm\sqrt{\ell\Delta x\Delta p}$, where the $\pm$ represents the fact that we do not know the sign. Looking closely it makes sense to assume that the $y_i$ should have a small bias $\mathcal{O}(\Delta x)$ if $\partial_x \rho(x,p) \neq 0$. Thus we assume the following (and similar for $[q]_{\alpha,\beta}$)

$$[y]_{\alpha,\beta} = \frac{\partial_x \rho(x_\alpha, p_\beta)}{\rho(x_\alpha, p_\beta)}\Delta x + \frac{1}{\ell\Delta x\Delta p}\zeta_{\alpha,\beta}, \tag{32}$$

where $\zeta_{\alpha,\beta} = \pm\mathcal{O}(1)$ is a random number (say a Gaussian) with mean 0 and variance $\mathcal{O}(1)$. Equation (32) should not be taken too literal: it only provides us with a reasonable estimate of the size of terms. In particular, there will be non-trivial prefactors.

This kind of argument can be made more concrete by explicitly averaging over the internal degrees of freedom $y_i, q_i$. For instance, we can assume that the $y_i$ in $(\alpha, \beta)$ are distributed according to the following distribution

$$f(y) = 1 + \Delta x \frac{\partial_x \rho(x_\alpha, p_\beta)}{\rho_{\alpha,\beta}} y, \tag{33}$$

taking into account the linear gradient term. Under this probability measure we find

$$\mathbb{E}[[y]_{\alpha,\beta}] = \frac{\Delta x}{12}\frac{\partial_x \rho(x_\alpha, p_\beta)}{\rho_{\alpha,\beta}} \sim \Delta x$$
$$\mathrm{Var}[[y]^2_{\alpha,\beta}] = \frac{1}{12 n_{\alpha,\beta}} \sim \frac{1}{\ell\Delta x\Delta p}. \tag{34}$$

The $q_i$'s coordinate can be treated similarly.

**Remark 1** *The scaling* (32) *does not require the central limit theorem, i.e. that all $y_i$ are iid. They can be non-indentical and even be corrrelated. In this case prefactors will obviously change, but the scalings will not be affected. The reason for this is large deviation theory: on an intuitive level it predicts that as $n_{\alpha,\beta} \to \infty$ the probability distribution of $[y]_{\alpha,\beta}$ can be approximated by a Gaussian around the largest value of the probability distribution. And this Gaussian will have expectation value and variance scaling as predicted by* (32). *Of course, there is no guarantee for that, but it is a reasonable guess (and also justified a posteriori by the agreement with numerical simulations). If in a particular case one expects a different scaling of $[y]_{\alpha,\beta}$ one can easily adjust the arguments.*

Proceeding now with either (32) or (34) and assuming that all fluid cells are independent, we find (we will use (34))

$$\mathbb{E}[\Upsilon_{\mathrm{FC}} - \Upsilon_{\mathrm{Micro}}] = \frac{\Delta x\Delta p}{12}\sum_{\alpha,\beta}\Big[ -\partial_x\Upsilon(x_\alpha, p_\beta)\partial_x\rho(x_\alpha, p_\beta)\Delta x^2$$

$$-\partial_p\Upsilon(x_\alpha, p_\beta)\partial_p\rho(x_\alpha, p_\beta)\Delta p^2 + \rho_{\alpha,\beta}\mathcal{O}(\Delta x^3)\Big] \tag{35}$$

$$\to -\frac{1}{12}\int dx\, dp\, \partial_x\Upsilon(x,p)\partial_x\rho(x,p)\Delta x^2 + \partial_p\Upsilon(x,p)\partial_p\rho(x,p)\Delta p^2 + \mathcal{O}(\Delta x^3) \tag{36}$$

and

$$\text{Var}[\Upsilon_{\text{FC}} - \Upsilon_{\text{Micro}}] = \frac{1}{12} \frac{\Delta x \Delta p}{\ell} \sum_{\alpha,\beta} \rho_{\alpha,\beta} \Big[ \partial_x \Upsilon(x_\alpha, p_\beta)^2 \Delta x^2 + \partial_p \Upsilon(x_\alpha, p_\beta)^2 \Delta p^2 + \mathcal{O}\big(\Delta x^3\big) \Big] \quad (37)$$

$$\to \frac{1}{12\ell} \int dx \, dp \, \rho(x,p) \Big[ \partial_x \Upsilon(x,p)^2 \Delta x^2 + \partial_p \Upsilon(x,p)^2 \Delta p^2 \Big] + \mathcal{O}\big(\Delta x^3 / \ell\big), \tag{38}$$

where in the last steps we took the continuum limit.

One way to think about this result is to introduce $\xi_{\text{FC}}^\Upsilon = \Upsilon_{\text{FC}} - \Upsilon_{\text{Micro}}$, which we can interpret as the emergent "noise" when describing the system via the coarse-grained $\rho_{\text{FC}}(x,p)$. From (36) and (38) we can read off that this noise has a mean and standard deviation scaling as

$$\mathbb{E}[\xi_{\text{FC}}^\Upsilon] \sim \Delta x^2 \qquad\qquad \text{stddev}[\xi_{\text{FC}}^\Upsilon] \sim \Delta x / \sqrt{\ell}. \tag{39}$$

It is natural to assume by that $\xi_{\text{FC}}^\Upsilon$ will be Gaussian and hence these two numbers are sufficient to describe its distribution. However, we have not investigated this noise as this is not the main point of our work.

Instead, we would like to point out the following interesting scaling behaviour (which we will encounter throughout this section): recall, that we start with a single fixed sample and coarse-grain it. Let us assume that $\Delta x \sim \ell^{\mu-1}$. Due to the above scalings we find that the error will scale as

$$\Upsilon_{\text{FC}} - \Upsilon_{\text{Micro}} \sim \max(\Delta x^2, \Delta x / \sqrt{\ell}) = \begin{cases} \ell^{-\frac{3}{2}+\mu} & \mu < 1/2 \\ \ell^{2\mu-2} & \mu > 1/2 \end{cases} \tag{40}$$

Hence, if we denote $\Upsilon_{\text{FC}} - \Upsilon_{\text{Micro}} \sim \ell^{-\nu}$ we find:

$$\nu = \begin{cases} \frac{3}{2} - \mu & \mu < 1/2 \\ 2 - 2\mu & \mu > 1/2. \end{cases} \tag{41}$$

This result is the theoretical line plotted in Fig. 2, where we also compare it to numerical simulations and find excellent agreement. Note that there is a "phase transition" in this scaling. For large fluid cells $\mu > 1/2$ the error is dominated by the systematic error of the Taylor expansion inside a fluid-cell. For small fluid cells $\mu < 1/2$ on the other hand the error is dominated by random fluctuations inside the fluid cells, i.e. it is a statistical error.

**Remark 2** *We want to stress again that the aim of this paper is to coarse-grain a single sample with large, but finite $N$. Hence there is no probability measure to actually take expectation values and variances over as we did in (36) and (38). Furthermore, the proposed distribution (34) is unphysical as it does not exclude hard rods from overlapping. Therefore, the use of a probability distribution is only to justify the extracted scalings, prefactors will obviously be incorrect.*

In this regard the more precise result of this section is that the error of coarse-graining a single configuration is given by

$$\Upsilon_{\text{FC}} - \Upsilon_{\text{Micro}} = \Delta x \Delta p \sum_{\alpha,\beta} \rho_{\alpha,\beta} \Big[ -\partial_x \Upsilon(x_\alpha, p_\beta)[y]_{\alpha,\beta} \Delta x - \partial_p \Upsilon(x_\alpha, p_\beta)[q]_{\alpha,\beta} \Delta p + \ldots \Big], \tag{42}$$

which for a sufficiently generic state scales as $\sim \max(\Delta x^2, \Delta x / \sqrt{\ell})$

**Remark 3** *Still, we believe that it should be possible to define a probability distribution on the $y_i$'s (at least as $n_{\alpha,\beta} \to \infty$) as $f(y) = \frac{1}{n_{\alpha,\beta}} \sum_{i \in C_{\alpha,\beta}} \delta(y - y_i)$. This idea might be worth investigating, as this might allow for much deeper understanding into the corrections to the present result.*

**Remark 4** *We would like to remark that these results hinge on the fact that the observable $\Upsilon(x, p)$ is smooth (or at least twice differentiable). If the observable is not smooth different scalings might occur.*

## 3.2 Contraction/Expansion

Next, let us understand the effect of the important mapping to contracted coordinates (and back to physical coordinates) in a similar fashion. Both of them behave similarly, and thus we only discuss the contraction mapping (4) in detail.

Using (11) the distribution of particles in contracted space $\hat{\rho}(\hat{x})$ is given as the push-forward $\hat{\rho}(\cdot, p) = \hat{X}_* \rho(\cdot, p)$. For measuring an observable $\hat{\Upsilon}(\hat{x}, p)$ this means:

$$\hat{\Upsilon} = \int \mathrm{d}x\, \mathrm{d}p\, \rho(x, p) \hat{\Upsilon}(\hat{X}(x), p). \tag{43}$$

Plugging the fluid cell coarse-grained density (19) into this, we obtain

$$\hat{\Upsilon}_{\mathrm{FC}} = \sum_{\alpha,\beta} \rho_{\alpha,\beta} \int_{x_\alpha - \Delta x/2}^{x_\alpha + \Delta x/2} \mathrm{d}x \int_{p_\beta - \Delta p/2}^{p_\beta + \Delta p/2} \mathrm{d}p\, \hat{\Upsilon}\left( x - a \sum_\beta \rho_\beta \int_{x_\beta - \Delta x/2}^{x_\beta + \Delta x/2} \mathrm{d}x'\, \theta(x - x') + \tfrac{a}{2\ell}, p \right) \tag{44}$$

$$= \Delta x \Delta p \sum_{\alpha,\beta} \rho_{\alpha,\beta} \int_{-1/2}^{1/2} \mathrm{d}y\, \mathrm{d}q\, \hat{\Upsilon}\Big( x_\alpha + y \Delta x$$

$$- a \Delta x \sum_{\alpha'} \bar{\rho}_{\alpha'} \int_{-1/2}^{1/2} \mathrm{d}y'\, \theta(x_\alpha - x_{\alpha'} + (y - y')\Delta x) + \tfrac{a}{2\ell}, p_\beta + q \Delta p \Big) \tag{45}$$

$$= \Delta x \Delta p \sum_\alpha \rho_{\alpha,\beta} \hat{\Upsilon}(\hat{x}_\alpha, p_\beta) - a \Delta x^2 \Delta p \sum_{\alpha,\beta,\alpha'} \rho_{\alpha,\beta} \bar{\rho}_{\alpha'} \partial_{\hat{x}} \hat{\Upsilon}(\hat{x}_\alpha, p_\beta)$$

$$\times \left( \int_{-1/2}^{1/2} \mathrm{d}y\, \mathrm{d}y'\, \theta(x_\alpha - x_{\alpha'} + (y - y')\Delta x) - \theta(x_\alpha - x_{\alpha'}) \right) + \mathcal{O}(\Delta x^2) \tag{46}$$

$$= \Delta x \Delta p \sum_\alpha \rho_{\alpha,\beta} \hat{\Upsilon}(\hat{x}_\alpha, p_\beta) \tag{47}$$

$$- a \Delta x^2 \Delta p \sum_{\alpha,\beta} \rho_{\alpha,\beta} \bar{\rho}_\alpha \partial_{\hat{x}} \hat{\Upsilon}(\hat{x}_\alpha, p_\beta) \underbrace{\int_{-1/2}^{1/2} \mathrm{d}y\, \mathrm{d}y'\, \mathrm{sgn}(y - y')}_{=0} + \mathcal{O}(\Delta x^2) \tag{48}$$

$$= \Delta x \Delta p \sum_\alpha \rho_{\alpha,\beta} \hat{\Upsilon}(\hat{x}_\alpha, p_\beta) + \mathcal{O}(\Delta x^2), \tag{49}$$

where we denoted

$$\bar{\rho}_\alpha = \Delta p \sum_\beta \rho_{\alpha,\beta} = \tfrac{1}{\ell \Delta x} \sum_\beta n_{\alpha,\beta} =: \tfrac{1}{\ell \Delta x} \bar{n}_\alpha \tag{50}$$

$$\hat{x}_\alpha = x_\alpha - a \Delta x \sum_{\alpha'} \bar{\rho}_{\alpha'} \theta(x_\alpha - x_{\alpha'}) + \tfrac{a}{2\ell} \tag{51}$$

and we defined $\theta(0) = 1/2$. In this computation we used that $\theta(x_\alpha - x_{\alpha'} + (y - y')\Delta x) - \theta(x_\alpha - x_{\alpha'})$ is only non-zero if $\alpha = \alpha'$.

On the other hand we have microscopically

$$\hat{\Upsilon}_{\text{Micro}} = \frac{1}{\ell} \sum_i \hat{\Upsilon}\left(x_i - \frac{a}{\ell}\sum_{j\neq i}\theta(x_i - x_j), p_i\right) \tag{52}$$

$$= \frac{1}{\ell}\sum_{\alpha,\beta}\sum_{i\in C_{\alpha,\beta}}\hat{\Upsilon}\Big(\hat{x}_\alpha + y_i\Delta x$$

$$- \frac{a}{\ell}\sum_{\alpha'}\sum_{j\in A_{\alpha'}}\theta(x_\alpha - x_{\alpha'} + (y_i - y_j)\Delta x) - \theta(x_\alpha - x_{\alpha'}), p_\beta + q_i\Delta p\Big) \tag{53}$$

$$= \frac{1}{\ell}\sum_{\alpha,\beta}\sum_{i\in C_{\alpha,\beta}}\hat{\Upsilon}\left(\hat{x}_\alpha + y_i\Delta x - \frac{a}{\ell}\sum_{j\in A_\alpha}\text{sgn}(y_i - y_j), p_\beta + q_i\Delta p\right) \tag{54}$$

and therefore

$$\hat{\Upsilon}_{\text{FC}} - \hat{\Upsilon}_{\text{Micro}} = -\frac{1}{\ell}\sum_{\alpha,\beta}\partial_{\hat{x}}\hat{\Upsilon}(\hat{x}_\alpha, p_\beta)\sum_{i\in C_{\alpha,\beta}}\left(y_i\Delta x - \frac{a}{\ell}\sum_{j\in A_\alpha}\text{sgn}(y_i - y_j)\right) \tag{55}$$

$$- \frac{\Delta p}{\ell}\sum_{\alpha,\beta}\partial_p\hat{\Upsilon}(\hat{x}_\alpha, p_\beta)\sum_{i\in C_{\alpha,\beta}}q_i + \mathcal{O}(\Delta x^2). \tag{56}$$

Note that using our convention we have $\text{sgn}(0) = \theta(0) - 1/2 = 0$.

Now we need to estimate the scaling of this expression. As in section 3.1 we will do this by assuming a distribution as (32) or (32). It is easy to see that $\mathbb{E}[\hat{\Upsilon}_{\text{FC}} - \hat{\Upsilon}_{\text{Micro}}] = \mathcal{O}(\Delta x^2)$. Therefore, let us focus on the variance. Since we assume $y_i$ and $q_i$ to be independent, the variance of (55-56) is just the variance of (55) plus the variance of (56). The latter scales, as in section 3.1, as $\Delta x^2/\ell$. Leveraging the independence of fluid cells, we have

$$\mathbb{E}[(\hat{\Upsilon}_{\text{FC}} - \hat{\Upsilon}_{\text{Micro}})^2] = \frac{1}{\ell^2}\sum_{\alpha,\beta,\beta'}\partial_{\hat{x}}\hat{\Upsilon}(\hat{x}_\alpha, p_\beta)\partial_{\hat{x}}\hat{\Upsilon}(\hat{x}_\alpha, p'_\beta)$$

$$\times \sum_{i\in C_{\alpha,\beta}}\sum_{i'\in C_{\alpha,\beta'}}\mathbb{E}\left[\left(y_i\Delta x - \frac{a}{\ell}\sum_{j\in A_\alpha}\text{sgn}(y_i - y_j)\right)\left(y_{i'}\Delta x - \frac{a}{\ell}\sum_{j'\in A_\alpha}\text{sgn}(y_{i'} - y_{j'})\right)\right]$$

$$+ \mathcal{O}(\Delta x^2/\ell) \tag{57}$$

$$= \frac{\Delta x^2}{\ell^2}\sum_{\alpha,\beta,}\partial_{\hat{x}}\hat{\Upsilon}(\hat{x}_\alpha, p_\beta)^2\sum_{i\in C_{\alpha,\beta}}\mathbb{E}[y_i^2]$$

$$- \frac{2a\Delta x}{\ell^3}\sum_{\alpha,\beta,\beta'}\partial_{\hat{x}}\hat{\Upsilon}(\hat{x}_\alpha, p_\beta)\partial_{\hat{x}}\hat{\Upsilon}(\hat{x}_\alpha, p'_\beta)\sum_{i\in C_{\alpha,\beta}}\sum_{i'\in C_{\alpha,\beta'}}\sum_{j'\in A_\alpha}\mathbb{E}[y_i\,\text{sgn}(y_{i'} - y_{j'})]$$

$$+ \frac{a^2}{\ell^4}\sum_{\alpha,\beta,\beta'}\partial_{\hat{x}}\hat{\Upsilon}(\hat{x}_\alpha, p_\beta)\partial_{\hat{x}}\hat{\Upsilon}(\hat{x}_\alpha, p'_\beta)\sum_{i\in C_{\alpha,\beta}}\sum_{i'\in C_{\alpha,\beta'}}\sum_{j,j'\in A_\alpha}\mathbb{E}[\text{sgn}(y_i - y_j)\,\text{sgn}(y_{i'} - y_{j'})]$$

$$+ \mathcal{O}(\Delta x^2/\ell). \tag{58}$$

All of these expectation values are only non-vanishing if at least two indices coincide, i.e. either $i = i'$, $i = j'$, etc. In this case each expectation value gives an $\mathcal{O}(1)$ contribution, hence we can estimate:

$$\mathbb{E}[(\hat{\Upsilon}_{\text{FC}} - \hat{\Upsilon}_{\text{Micro}})^2] = \mathcal{O}(\Delta x^2/\ell). \tag{59}$$

This estimate was obtain by simply counting the number of non-zero summands in (58) and using $|C_{\alpha,\beta}| = n_{\alpha,\beta} \sim \ell\Delta x\Delta p$, $|A_\alpha| \sim \ell\Delta x$ and that the sums over $\alpha$ run over $\sim 1/\Delta x$ summands (and similarly sums over $\beta$ run over $\sim 1/\Delta p$ summands).

Hence, we conclude

$$\mathbb{E}[\hat{\Upsilon}_{\text{FC}} - \hat{\Upsilon}_{\text{Micro}}] \sim \Delta x^2 \tag{60}$$

$$\text{Var}[\hat{\Upsilon}_{\text{FC}} - \hat{\Upsilon}_{\text{Micro}}] = \frac{\Delta x^2}{\ell}. \tag{61}$$

These are the same scalings as in (40) and hence the error $\Upsilon_{\text{FC}} - \Upsilon_{\text{Micro}} \sim \ell^{-\nu}$ scales again as:

$$\nu = \begin{cases} \frac{3}{2} - \mu & \mu < 1/2 \\ 2 - 2\mu & \mu > 1/2. \end{cases} \tag{62}$$

**Remark 5** *Note that the additional constant $\frac{a}{2\ell}$ from (11) cancels exactly with the extension of the sum $\sum_{i \neq j} \to \sum_j$ in the step going from (52) to (53). Hence, this choice of constant in crucial.*

### 3.3 Non-interacting time evolution

Now we consider the case where we let the system evolve with the non-interacting time-evolution, which in the hard rods happens in contracted coordinates.

For now let us study the simplified case of non-interacting particles $a \to 0$, where the evolution is simply given by:

$$x_i \to x_i + p_i t. \tag{63}$$

This means that the observable is given by

$$\Upsilon(t)_{\text{Micro}} = \frac{1}{\ell} \sum_i \Upsilon(x_i + p_i t, p_i) \tag{64}$$

or in continuous form by

$$\Upsilon(t)_{\text{Smooth}} = \int dx \, dp \, \rho(x, p) \Upsilon(x + pt, p). \tag{65}$$

Note that (for now) we will assume that we study Euler times, i.e. $t = \mathcal{O}(1)$ is a fixed number. However, we will keep time explicit for later discussion.

Under fluid cell coarse-graining (19) we simply find

$$\Upsilon(t)_{\text{FC}} = \sum_{\alpha, \beta} \rho_{\alpha, \beta} \int_{x_\alpha - \Delta x/2}^{x_\alpha + \Delta x/2} dx \int_{p_\beta - \Delta p/2}^{p_\beta + \Delta p/2} dp \, \Upsilon(x + pt, p) \tag{66}$$

$$= \Delta x \Delta p \sum_{\alpha, \beta} \rho_{\alpha, \beta} \int_{-1/2}^{1/2} dy \, dq \, \Upsilon(x_\alpha + p_\beta t + y \Delta x + qt \Delta p, p_\beta + q \Delta p) \tag{67}$$

$$= \Delta x \Delta p \sum_{\alpha, \beta} \rho_{\alpha, \beta} \Upsilon(x_\alpha + p_\beta t, p_\beta) + \frac{1}{24} \Delta x \Delta p \sum_{\alpha, \beta} \rho_{\alpha, \beta}$$

$$\times \left[ \partial_x^2 \Upsilon(x_\alpha + p_\beta t, p_\beta)(\Delta x^2 + t^2 \Delta p^2) + \partial_p^2 \Upsilon(x_\alpha + p_\beta t, p_\beta) \Delta p^2 \right] + \mathcal{O}\left( \Delta x^3 + (t \Delta p)^3 \right) \tag{68}$$

$$= \Delta x \Delta p \sum_{\alpha, \beta} \rho_{\alpha, \beta} \Upsilon(x_\alpha + p_\beta t, p_\beta) + \mathcal{O}\left( \Delta x^2 + (t \Delta p)^2 \right). \tag{69}$$

On the other hand we also have

$$\Upsilon(t)_{\text{Micro}} = \frac{1}{\ell} \sum_{\alpha,\beta} \sum_{i \in C_{\alpha,\beta}} \Upsilon(x_i + p_i t, p_i) \tag{70}$$

$$= \frac{1}{\ell} \sum_{\alpha,\beta} \sum_{i \in C_{\alpha,\beta}} \Upsilon(x_\alpha + p_\beta t + y_i \Delta x + q_i t \Delta p, p_\beta + q_i \Delta p) \tag{71}$$

$$= \Delta x \Delta p \sum_{\alpha,\beta} \rho_{\alpha,\beta} \Upsilon(x_\alpha + p_\beta t, p_\beta) + \Delta x \Delta p \sum_{\alpha,\beta} \rho_{\alpha,\beta}$$
$$\times \left[ \partial_x \Upsilon(x_\alpha + p_\beta t, p_\beta)([y]_{\alpha,\beta} \Delta x + [q]_{\alpha,\beta} t \Delta p) + \partial_p \Upsilon(x_\alpha + p_\beta t, p_\beta)[q]_{\alpha,\beta} \Delta p \right]$$
$$+ \mathcal{O}\big(\Delta x^2 + (t \Delta p)^2\big). \tag{72}$$

Taking the difference and applying the same resaoning as in section 3.1, we find that

$$\mathbb{E}[\Upsilon(t)_{\text{FC}} - \Upsilon(t)_{\text{Micro}}] \sim \Delta x^2 + t \Delta p^2 \tag{73}$$

$$\text{Var}[\Upsilon(t)_{\text{FC}} - \Upsilon(t)_{\text{Micro}}] \sim \frac{\Delta x^2}{\ell} + \frac{t^2 \Delta p^2}{\ell} \tag{74}$$

which, for $t = \mathcal{O}(1)$, is again the same scaling as in (40), i.e. $\Upsilon(t)_{\text{FC}} - \Upsilon(t)_{\text{Micro}} \sim \ell^{-\nu}$ with

$$\nu = \begin{cases} \frac{3}{2} - \mu & \mu < 1/2 \\ 2 - 2\mu & \mu > 1/2. \end{cases} \tag{75}$$

## 3.4 The full evolution

Now, let us study the full evolution: using fluid cell coarse-graining the value of the observable $\Upsilon(x, p)$ at time $t$ is given by

$$\Upsilon(t)_{\text{FC}}^{\text{HR}} = \int dx \, dp \, \rho_{\text{FC}}(x, p) \Upsilon(X_{\text{FC}}(t, x, p), p), \tag{76}$$

where

$$X_{\text{FC}}(t, x, p) = \hat{X}_{\text{FC}}(x) + pt + a \int dx' \, dp' \, \rho_{\text{FC}}(y, q) \theta(\hat{X}_{\text{FC}}(x) + pt - \hat{X}_{\text{FC}}(x') - p't) - \frac{a}{2\ell} \tag{77}$$

$$\hat{X}_{\text{FC}}(x) = x - a\Phi_{\text{FC}}(x_{\alpha(x)}) - a\bar{\rho}_{\alpha(x)}(x - x_{\alpha(x)}) + \frac{a}{2\ell} \tag{78}$$

$$\Phi_{\text{FC}}(x_\alpha) = \Delta x \sum_{\alpha' < \alpha} \bar{\rho}_{\alpha'} + \bar{\rho}_\alpha \Delta x / 2 \tag{79}$$

and $\alpha(x)$ is the fluid cell in which $x$ is. Using similar arguments as before we find

$$\Upsilon(t)_{\text{FC}}^{\text{HR}} = \Delta x \Delta p \sum_{\alpha,\beta} \rho_{\alpha,\beta} \Upsilon(x_{\alpha,\beta}(t), p_\beta) + \mathcal{O}\big(\Delta x^2\big), \tag{80}$$

where

$$x_{\alpha,\beta}(t) = \hat{x}_\alpha + p_\beta t + a\Delta x \Delta p \sum_{\alpha',\beta'} \rho_{\alpha',\beta'} \theta(\hat{x}_\alpha + p_\beta t - \hat{x}_{\alpha'} - p_{\beta'} t) - \frac{a}{2\ell} \tag{81}$$

Here, for simplicity we assume that $t = \mathcal{O}(1)$, s.t. $\Delta p t \sim \Delta x$.

The microscopic evolution is given by

$$\Upsilon(t)_{\text{Micro}}^{\text{HR}} = \frac{1}{\ell} \sum_i \Upsilon(x_i(t), p_i) \tag{82}$$

and thus

$$\Upsilon(t)_{\text{FC}}^{\text{HR}} - \Upsilon(t)_{\text{Micro}}^{\text{HR}} = -\frac{1}{\ell}\sum_{\alpha,\beta}\partial_x\Upsilon(x_{\alpha,\beta}(t),p_\beta)\left(\sum_{i\in C_{\alpha,\beta}}x_i(t)-x_{\alpha,\beta}(t)\right)$$

$$-\frac{\Delta p}{\ell}\sum_{\alpha,\beta}\partial_p\Upsilon(x_{\alpha,\beta}(t),p_\beta)\left(\sum_{i\in C_{\alpha,\beta}}q_i\right)+\mathcal{O}(\Delta x^2). \tag{83}$$

In the following denote by

$$\hat{y}_i = \frac{\hat{x}_i - \hat{X}_{\text{FC}}(x_\alpha)}{\Delta x} \tag{84}$$

and observe

$$x_i(t) - x_{\alpha,\beta}(t) = \hat{y}_i\Delta x + q_i\Delta pt$$
$$+\frac{a}{\ell}\sum_j\theta(\hat{x}_i+p_it-\hat{x}_j-p_jt)-\theta(\hat{x}_\alpha+p_\beta t-\hat{x}_{\alpha(x_j)}-p_{\beta(x_j)}t). \tag{85}$$

Defining $z_{\alpha,\beta,\alpha',\beta'} = \hat{x}_\alpha + p_\beta t - \hat{x}_{\alpha(x_j)} - p_{\beta(x_j)}t$, this gives

$$x_i(t) - x_{\alpha,\beta}(t) = \hat{y}_i\Delta x + q_i\Delta pt + \frac{a}{\ell}\sum_{\alpha',\beta'}\sum_{j\in C_{\alpha',\beta'}}$$

$$\times\theta(z_{\alpha,\beta,\alpha',\beta'}+(\hat{y}_i-\hat{y}_j)\Delta x+(q_i-q_j)\Delta pt)-\theta(z_{\alpha,\beta,\alpha',\beta'}) \tag{86}$$

$$= \hat{y}_i\Delta x + q_i\Delta pt + \frac{a}{\ell}\sum_{\alpha',\beta'}\sum_{j\in C_{\alpha',\beta'}}$$

$$\times\theta(0<-\text{sgn}(y_{ij}(t))z_{\alpha,\beta,\alpha',\beta'}<|y_{ij}(t)|\,\text{sgn}(y_{ij}(t))). \tag{87}$$

Here $y_{ij}(t) = (\hat{y}_i-\hat{y}_j)\Delta x + (q_i-q_j)\Delta pt$ and we used

$$\theta(x+y)-\theta(x) = \theta(0<-\text{sgn}(y)x<|y|)\,\text{sgn}(y). \tag{88}$$

Next, we need to estimate the scaling of this. Unfortunately, performing the averaging (32) is significantly more complicated now. However, recall that the probability measure (32) is not the actual microscopic randomness in the model. Instead, it is meant to guide us towards the scaling of these terms. Therefore, let us use what we learned from previous sections to try to estimate the size of (83): first, due to the $\theta(0<-\text{sgn}(y_{ij}(t))z_{\alpha,\beta,\alpha',\beta'}<|y_{ij}(t)|\,\text{sgn}(y_{ij}(t))$ the sum will only give a contribution if $z_{\alpha,\beta,\alpha',\beta'} = \mathcal{O}(\Delta x)$. Therefore, for each $\alpha,\beta,\beta'$, there will be only a few $\alpha'$ giving a contribution. Thus the sum has $\sim \ell\Delta x^2/\Delta x^3 = \ell/\Delta x$ terms. On average, each one of these terms averages to 0, because the sign of the $y_{ij}$ should be fully random. Thus, the expectation value of (83) is of order $\Delta x^2$, as before. However, the variance of each $\text{sgn}(y_{ij})$ is of $\mathcal{O}(1)$, implying that the variance of (87) is of order $\mathcal{O}(\Delta x^2)$. Therefore, we expect that the expectation value and variance of (83) again scale as

$$\mathbb{E}[\Upsilon(t)_{\text{FC}}^{\text{HR}} - \Upsilon(t)_{\text{Micro}}^{\text{HR}}] \sim \Delta x^2 \tag{89}$$

$$\text{Var}[\Upsilon(t)_{\text{FC}}^{\text{HR}} - \Upsilon(t)_{\text{Micro}}^{\text{HR}}] \sim \frac{N}{\ell^2}\text{Var}[x_i(t)-x_{\alpha,\beta}(t)] \sim \Delta x^2/\ell \tag{90}$$

which means that, as in previous sections, the error scales as $\Upsilon(t)_{\text{FC}}^{\text{HR}} - \Upsilon(t)_{\text{Micro}}^{\text{HR}} \sim \ell^{-\nu}$ with

$$\nu = \begin{cases} \frac{3}{2}-\mu & \mu<1/2 \\ 2-2\mu & \mu>1/2. \end{cases} \tag{91}$$

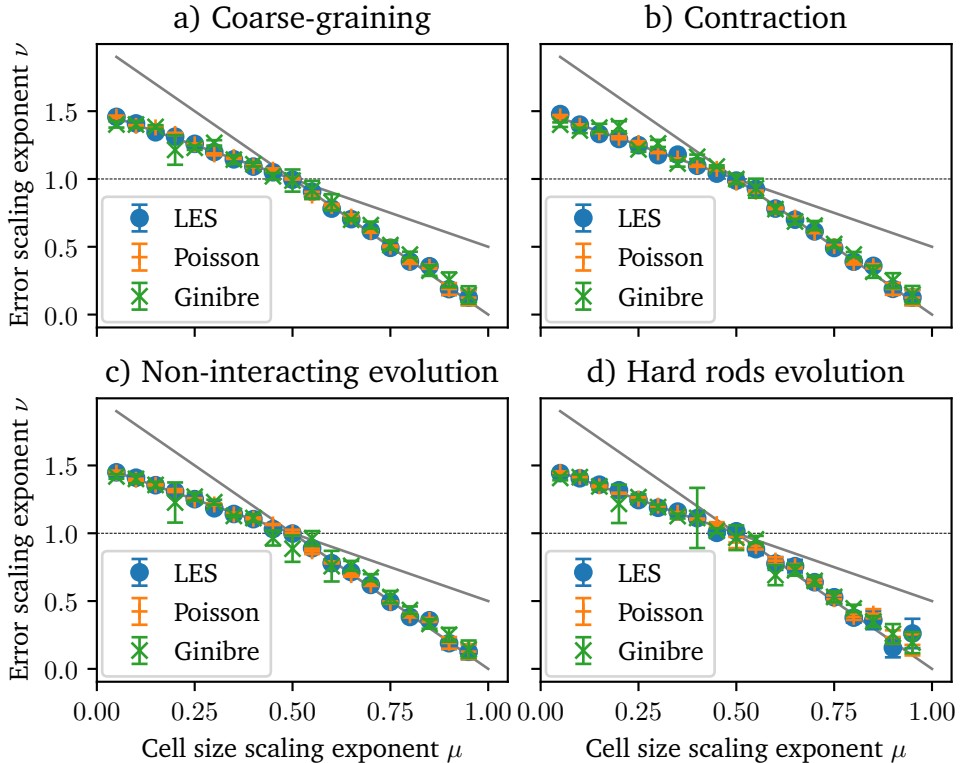

Figure 2: Numerically extracted scaling of the error $\xi \sim \ell^{-\nu}$ as function of coarse-graining scale $\Delta x, \Delta p \sim \ell^{\mu-1}$ for fluid cell coarse-graining. We dististinguish the four approximations discussed in the text: a) coarse-graining of the initial configuration (section 3.1), b) contracting map (section 3.2), c) non-interacting time evolution (section 3.3) and d) the full hard rods time evolution (section 3.4). The points are the numerical results for the three ensembles discussed in section 3.5. The errorbars represent the fit error on $\nu$ (note that the error for the Ginibre states are larger, since they had to be obtained on smaller system sizes). The numerical results agree well with the analytical predictions obtained in this section. The dashed line at $\nu = 1$ is diffusive scaling. If $\nu > 1$, Euler hydrodynamics is accurate also on the diffusive scale. Further details of the numerical simulations are explained in appendix B.

## 3.5 Numerical simulations

Verifying scalings such as (91) is not so trivial, because the error is only asymptotically: for a given fixed configuration the error might be very large. To extract any meaningful scalings we thus average the error over many samples drawn from appropriate probability distributions $\mathbb{E}[\dots]$. To be precise we compute

$$\xi = \sqrt{\mathbb{E}[(\Upsilon_{\text{FC}} - \Upsilon_{\text{Micro}})^2]} \tag{92}$$

$$= \sqrt{(\mathbb{E}[\Upsilon_{\text{FC}} - \Upsilon_{\text{Micro}}])^2 + \text{Var}[\Upsilon_{\text{FC}} - \Upsilon_{\text{Micro}}]} \tag{93}$$

for different $\ell \to \infty$, with $\Delta x, \Delta p \sim \ell^{\mu-1}$, and then extract its scaling $\xi \sim \ell^{-\nu}$ using a numerical fit of the form $\xi = a\ell^{-\nu}$. Here $0 < \mu < 1$ controls the coarse-graining scale. To be more precise, we do the fit on the expectation value and the standard deviation separately and then take the minimum of the obtained $\nu$. This is to avoid transition effects around $\mu = 1/2$, where the behaviour shifts from variance dominated to expectation value dominated.

To demonstrate the universality of the scalings we use different measures $\mathbb{E}[\dots]$, described in the following:

- Local equilibrium state (LES): This is the canonical initial state (2) considered in hydro-dynamics. We choose a target density $\rho(x,p) = \frac{A}{2\pi}e^{-\frac{1}{2}(x^2+p^2)}$. Note that such states can be generated quite efficiently, see e.g. [18].

- Poisson point process (Poisson): We generate a Poisson point process in the contracted coordinates $\hat{x}$ with target density $\hat{\rho}(\hat{x},p) = \frac{A}{2\pi}e^{-\frac{1}{2}(\hat{x}^2+p^2)}$ and then expand the config-uration to physical coordinates $\hat{x} \to x$ to obtain a valid hard rods configuration. Note that the resulting probability measure will have non-trivial long range correlations.

- Ginibre ensemble (Ginibre): We first draw the number of particles $N \sim \text{Pois}(A\ell)$ and then sample a random Ginibre matrix $\mathbf{Z} = (\mathbf{X} + i\mathbf{Y})/\sqrt{2N}$ (here $\mathbf{X}, \mathbf{Y}$ are matrices filled with iid standard Gaussians). Then, we compute the eigenvalues $z_i$ of this matrix and identify it with a particle $(\hat{x}_i, p_i)$ in contracted space via $\hat{x}_i + ip_i = f(|z_i|)z_i$, where $f(r) = \frac{1}{r}\sqrt{-2\log(1-r^2)}$. Note that since the eigenvalues of the Ginibre ensemble are distributed uniformly in the disc $|z| < 1$ [50], the rescaling factor $f(r)$ makes $(\hat{x}_i, p_i)$ distributed as $\hat{\rho}(\hat{x},p) = \frac{A}{2\pi}e^{-\frac{1}{2}(\hat{x}^2+p^2)}$. We then expand this configuration to physical coordinates.

In this section we choose the amplitude $A = 5$. Note that the first two ensembles were chosen to be physical states of hard rods. The last one (Ginibre) was chosen on purpose to be as unphysical as possible. The "eigenvalue repulsion" property of random matrices leads to local correlations of the form $\langle \delta\hat{\rho}(\hat{x},p)\delta\hat{\rho}(\hat{y},q)\rangle \sim (\delta''(\hat{x}-\hat{y})\delta(p-q) + \delta(\hat{x}-\hat{y})\delta''(p-q))$ [51], which is clearly fundamentally different from the usual Poisson process like correlations of hard rod local GGE states in contracted coordinates $\langle \delta\hat{\rho}(\hat{x},p)\delta\hat{\rho}(\hat{y},q)\rangle \sim \delta(\hat{x}-\hat{y})\delta(p-q)$. Note that because the evolution in contracted coordinates is trivial, it is easy to see that the (singular parts of the) correlations of the (Ginibre) states will remain intact. Hence, they will never become GGE like, or in other words: (Ginibre) states never thermalize to a GGE, not even locally(!). Therefore, the (Ginibre) states extremely unphysical states.

The numerically extracted scalings are found in Fig. 2, which all agree with the analytically predicted scalings.

**Remark 6** *The fact that GHD also works for the unphysical (Ginibre) states clearly demonstrate that the local equilibrium assumption of hydrodynamics is not at all required. Instead, GHD emerges already if particles are sufficiently generically placed in fluid cells. Note that even if we do not assume this, the correction term (83) is of order $\mathcal{O}(\Delta x)$ as long as the density is bounded. Thus GHD should emerge even more generally (however, with worse scaling of the error). It would be interesting to check in non-integrable models whether or not hydrodynamics applies to non-physical states as well.*

**Remark 7** *Again, we would like to emphasise that the use of probability measures as random initial states is purely used to extract the scaling in a reliable way. Note that what we show here is $\sqrt{\mathbb{E}[(\Upsilon_{\text{FC}} - \Upsilon_{\text{Micro}})^2]} \to 0$, which means that Euler GHD works almost surely for all states in the ensemble $\mathbb{E}[\ldots]$. This is much stronger than showing GHD in the usual sense, which would mean $\mathbb{E}[\Upsilon_{\text{FC}}] \to \mathbb{E}[\Upsilon_{\text{Micro}}]$. Unfortunately, as $\ell \to \infty$ all of the ensembles $\mathbb{E}[\ldots]$ only cover a very tiny subset of all possible initial states. This is why our analytical analysis is so important: it predicts that GHD should work almost surely for any (reasonable) ensemble of states.*

## 4  Coarse-graining 2: Smoothing

We would like to present alternative common way to approximate a discrete probability distribution by a smooth one. The idea is to replace the $\delta$ peaks in (14) by smooth functions

$$\delta(x)\delta(p) \to \tfrac{1}{\Delta x \Delta p}\eta(\tfrac{x}{\Delta x}, \tfrac{p}{\Delta p}). \tag{94}$$

Here $\eta(x,p)$ is a non-negative smooth function that is symmetric in $x$ and $p$ and integrates to $\int \mathrm{d}x\,\mathrm{d}p\,\eta(x,p) = 1$. For convenience, we will also assume

$$\int \mathrm{d}x\,\mathrm{d}p\,\eta(x,p)x^2 = \int \mathrm{d}x\,\mathrm{d}p\,\eta(x,p)p^2 = 1. \tag{95}$$

A simple example of this is a Gaussian

$$\eta(x,p) = \tfrac{1}{2\pi}e^{-\frac{x^2+p^2}{2}}, \tag{96}$$

but many more functions are available. As before $\Delta x$ and $\Delta p$ control the width of the smoothening window. We therefore define

$$\rho_{\mathrm{Smooth}}(x,p) = \tfrac{1}{\ell}\sum_i \tfrac{1}{\Delta x \Delta p}\eta(\tfrac{x-x_i}{\Delta x}, \tfrac{p-p_i}{\Delta p}), \tag{97}$$

which is also the convolution of $\rho_{\mathrm{Micro}}$ by (94).

### 4.1  Initial state coarse-graining

Integrating this against an observable we have

$$\Upsilon_{\mathrm{Smooth}} = \int \mathrm{d}x\,\mathrm{d}p\,\rho_{\mathrm{Smooth}}(x,p)\Upsilon(x,p) = \tfrac{1}{\ell}\sum_i \int \mathrm{d}y\,\mathrm{d}q\,\eta(y,q)\Upsilon(x_i + y\Delta x, p_i + q\Delta p) \tag{98}$$

$$= \tfrac{1}{\ell}\sum_i \Upsilon(x_i, p_i) + \tfrac{1}{2\ell}\sum_i \partial_x^2 \Upsilon(x_i, p_i)\left[\int \mathrm{d}y\,\mathrm{d}q\,\eta(y,q)y^2\right]\Delta x^2 \tag{99}$$

$$+ \tfrac{1}{2\ell}\sum_i \partial_p^2 \Upsilon(x_i, p_i)\left[\int \mathrm{d}y\,\mathrm{d}q\,\eta(y,q)q^2\right]\Delta p^2 + \mathcal{O}(\Delta x^3). \tag{100}$$

Hence, we find that the difference is given by

$$\Upsilon_{\mathrm{Smooth}} - \Upsilon_{\mathrm{Micro}} = \tfrac{1}{2\ell}\sum_i \partial_x^2 \Upsilon(x_i, p_i)\Delta x^2 + \tfrac{1}{2\ell}\sum_i \partial_p^2 \Upsilon(x_i, p_i)\Delta p^2 + \mathcal{O}(\Delta x^3) \tag{101}$$

$$\to \tfrac{1}{2}\int \mathrm{d}x\,\mathrm{d}p\,\rho(x,p)(\partial_x^2 \Upsilon(x,p)\Delta x^2 + \partial_p^2 \Upsilon(x,p)\Delta p^2). \tag{102}$$

In the last step we took a continuum limit. The error of the approximation for $\Delta x, \Delta p \sim \ell^{\mu-1}$ is $\ell^{-\nu}$ with

$$\nu = 2 - 2\mu. \tag{103}$$

For large $\mu > 1/2$ this approximation is therefore as good as the fluid cell one, but for smaller $\mu < 1/2$ it is better. This is due to the smoothness of the approximation.

## 4.2 Contraction/Expansion

Next, we will study how contraction behaves under the smoothing coarse-graining. Let us denote $\bar{\eta}(x) = \int \mathrm{d}p\, \eta(x, p)$. From (11) we have

$$\hat{X}_{\text{Smooth}}(x) = x - \frac{a}{\ell} \sum_i \int \mathrm{d}y\, \bar{\eta}(y)\theta(x - x_i - y\Delta x) + \frac{a}{2\ell}. \tag{104}$$

We can compute the observable $\hat{\Upsilon}$ as

$$\hat{\Upsilon}_{\text{Smooth}} = \frac{1}{\ell} \sum_i \int \mathrm{d}y\, \mathrm{d}q\, \eta(y, q)$$

$$\times \hat{\Upsilon}\left( x_i + y\Delta x - \frac{a}{\ell} \sum_j \int \mathrm{d}y'\, \bar{\eta}(y')\theta(x_i - x_j + (y - y')\Delta x) + \frac{a}{2\ell}, p_i + q\Delta p \right). \tag{105}$$

We find that

$$\hat{\Upsilon}_{\text{Smooth}} - \hat{\Upsilon}_{\text{Micro}} = \frac{1}{\ell} \sum_i \partial_{\hat{x}} \hat{\Upsilon}(\hat{x}_i, p_i)\left[ -\frac{a}{\ell} \sum_j \int \mathrm{d}y\, \mathrm{d}y'\, \bar{\eta}(y)\bar{\eta}(y') \right.$$

$$\left. \times \theta(x_i - x_j + (y - y')\Delta x) + \frac{a}{2\ell} + \frac{a}{\ell} \sum_{j \neq i} \theta(x_i - x_j) \right] + \mathcal{O}(\Delta x^2) \tag{106}$$

Let us define the function

$$f(x) = \int \mathrm{d}y\, \mathrm{d}y'\, \bar{\eta}(y)\bar{\eta}(y')\theta(x + y - y') - \theta(x). \tag{107}$$

Note that this function is anti-symmetric and vanishes as $x \to \pm\infty$:

$$f(-x) = -f(x) \qquad\qquad \lim_{x \to \pm\infty} f(x) = 0. \tag{108}$$

We can use this to write

$$\hat{\Upsilon}_{\text{Smooth}} - \hat{\Upsilon}_{\text{Micro}} = \frac{1}{\ell} \sum_i \partial_{\hat{x}} \hat{\Upsilon}(\hat{x}_i, p_i)\left[ -\frac{a}{\ell} \sum_{j \neq i} f\left(\frac{x_i - x_j}{\Delta x}\right) \right.$$

$$\left. + \frac{a}{\ell}\left( \frac{1}{2} - \int \mathrm{d}y\, \mathrm{d}y'\, \eta(y)\eta(y)\theta(\eta - \eta') \right) \right] + \mathcal{O}(\Delta x^2) \tag{109}$$

$$= -\frac{a}{\ell^2} \sum_{i \neq j} \partial_{\hat{x}} \hat{\Upsilon}(\hat{x}_i, p_i) f\left(\frac{x_i - x_j}{\Delta x}\right) + \mathcal{O}(\Delta x^2). \tag{110}$$

Note that terms in this sum will only contribute if $x_i - x_j = \mathcal{O}(\Delta x)$, meaning that for each $i$, $j$ runs over $\sim \Delta x \ell$ terms. Unlike in the case of fluid cell coarse-graining, we cannot average over the microscopic randomness to analytically estimate the size of the error. We can still obtain a sensible estimate based on the heuristic understanding we gained in section 3. First, note that if $x_i - x_j = \mathcal{O}(\Delta x)$ then $f((x_i - x_j)/\Delta x)$ is an $\mathcal{O}(1)$ number with an arbitrary sign (recall that $f(x)$ is antisymmetric). As a crude estimate we can assume that all of these numbers are iid, which means that the total sum averages to 0 and has variance $\sim \frac{N\Delta x}{\ell^4}\mathcal{O}(1) = \mathcal{O}(\Delta x/\ell^2)$.

Thus, we conclude that

$$\hat{\Upsilon}_{\text{Smooth}} - \hat{\Upsilon}_{\text{Micro}} \sim \max\left( \Delta x^2, \frac{\sqrt{\Delta x}}{\ell} \right), \tag{111}$$

which scales as $\ell^{-\nu}$, with

$$\nu = \begin{cases} 2 - 2\mu & \mu > 1/3 \\ \frac{3}{2} - \frac{\mu}{2} & \mu < 1/3. \end{cases} \tag{112}$$

Note that this is is a different scaling from (40) and (103). It again shows the "phase transition" behaviour, but at a different point.

**Remark 8** *Note that also with the smoothing the additional constant $\frac{a}{2\ell}$ in (11) is needed to cancel a term in the derivation (the second term in (109))*

### 4.3 Non-interacting time evolution

The non-interacting time evolution after smoothing is given by

$$\Upsilon(t)_{\text{Smooth}} = \frac{1}{\ell} \sum_i \int \mathrm{d}y \, \mathrm{d}q \, \eta(y, q) \Upsilon(x_i + p_i t + y\Delta x + qt\Delta p, p_i + q\Delta p) \tag{113}$$

$$= \frac{1}{\ell} \sum_i \Upsilon(x_i + p_i t + y\Delta x + qt\Delta p, p_i + q\Delta p). \tag{114}$$

Hence we find

$$\Upsilon(t)_{\text{Smooth}} - \Upsilon(t)_{\text{Micro}} = \frac{1}{2\ell} \sum_i \partial_x^2 \Upsilon(x_i + p_i t, p_i)\big(\Delta x^2 + (t\Delta p)^2\big)$$

$$+ \partial_p^2 \Upsilon(x_i + p_i t, p_i)\Delta p^2 + \mathcal{O}\big(\Delta x^3 + (t\Delta p)^3\big) \tag{115}$$

$$\to \frac{1}{2} \int \mathrm{d}x \, \mathrm{d}p \, \rho(x, p)\Big(\partial_x^2 \Upsilon(x + pt, p)\big(\Delta x^2 + (t\Delta p)^2\big)$$

$$+ \partial_p^2 \Upsilon(x + pt, p)\Delta p^2\Big) + \mathcal{O}\big(\Delta x^3 + (t\Delta p)^3\big), \tag{116}$$

from which we conclude

$$\Upsilon(t)_{\text{Smooth}} - \Upsilon(t)_{\text{Micro}} \sim \Delta x^2 + (t\Delta p)^2. \tag{117}$$

For $t = \mathcal{O}(1)$, this is again the same scaling as in (103). Thus error of the approximation scales as $\ell^{-\nu}$ with

$$\nu = 2 - 2\mu. \tag{118}$$

### 4.4 Full time evolution

With the explicit GHD trajectories after smoothing

$$X_{\text{Smooth}}(t, x, p) = \hat{X}_{\text{Smooth}}(x) + pt - \frac{a}{2\ell}$$

$$+ a \int \mathrm{d}x' \, \mathrm{d}p' \, \rho_{\text{Smooth}}(y, q)\theta(\hat{X}_{\text{Smooth}}(x) + pt - \hat{X}_{\text{Smooth}}(x') - p't), \tag{119}$$

where

$$\hat{X}_{\text{Smooth}}(x) = x - \frac{a}{\ell} \sum_k \int \mathrm{d}y \, \mathrm{d}q \, \eta(y, q)\theta(x - x_k - y\Delta x) + \frac{a}{2\ell} \tag{120}$$

the value of an observable $\Upsilon(x, p)$ at time $t$ after smoothing is given by

$$\Upsilon(t)_{\text{Smooth}}^{\text{HR}} = \int dx \, dp \, \rho_{\text{Smooth}}(x, p) \Upsilon(X_{\text{Smooth}}(t, x, p), p) \tag{121}$$

$$= \frac{1}{\ell} \sum_i \int dy \, dq \, \eta(y, q) \Upsilon(\hat{X}_{\text{Smooth}}(x_i + y \Delta x)$$

$$+ \frac{a}{\ell} \sum_j \int dy' \, dq' \, \eta(y', q') \theta(\hat{X}_{\text{Smooth}}(x_i + y \Delta x) - \hat{X}_{\text{Smooth}}(x_j + y' \Delta p)$$

$$+ (p_i - p_j)t + (q - q') \Delta p t), p_i + q \Delta p = . \tag{122}$$

Using similar arguments as before we find

$$\Upsilon(t)_{\text{Smooth}}^{\text{HR}} - \Upsilon(t)_{\text{Micro}}^{\text{HR}} = \frac{1}{\ell} \sum_i \partial_x \Upsilon(x_i(t), p_i) \left[ -\frac{a}{\ell} \sum_{j \neq i} f\left(\frac{x_i - x_j}{\Delta x}\right) + \frac{a}{\ell} \sum_{j \neq i} g(x_i, p_i; x_j, p_j) \right]$$

$$+ \mathcal{O}(\Delta x^2). \tag{123}$$

Here we defined

$$f(x) = \int dy \, dy' \, \eta(y) \eta(y') (\theta(x + y - y') - \theta(x)) \tag{124}$$

$$g(x_i, p_i; x_j, p_j) = \int dy \, dq \, dy' \, dq' \, \eta(y, q) \eta(y', q') \left[ \theta(\hat{X}_{\text{Smooth}}(x_i + y \Delta x) - \hat{X}_{\text{Smooth}}(x_j + y' \Delta p) \right.$$

$$\left. + (p_i - p_j)t + (q - q') \Delta p t) - \theta(\hat{x}_i - \hat{x}_j + (p_i - p_j)t) \right]. \tag{125}$$

Note that $g(x_i, p_i; x_j, p_j) = -g(x_j, p_j; x_i, p_i)$ and $g(x_i, p_i; x_j, p_j)$ is only non-negligible if $\hat{x}_i + p_i t - \hat{x}_j - p_j t = \mathcal{O}(\Delta x + \Delta p t)$. Therefore, as in section 4.2, the terms in the bracket in (123) will be of $\mathcal{O}(1)$ but with a fluctuating sign. Thus, following the same reasoning as in section 4.2, we arrive at the same result

$$\Upsilon(t)_{\text{Smooth}}^{\text{HR}} - \Upsilon(t)_{\text{Micro}}^{\text{HR}} \sim \max(\Delta x^2, \frac{\sqrt{\Delta x}}{\ell}), \tag{126}$$

which scales as $\ell^{-\nu}$, with

$$\nu = \begin{cases} 2 - 2\mu & \mu > 1/3 \\ \frac{3}{2} - \frac{\mu}{2} & \mu < 1/3. \end{cases} \tag{127}$$

## 4.5   Numerical simulations

As for the case of fluid cell averaging we also compare the results with numerical simulations and find good agreement, see Fig. 3. Here we used the same initial state ensembles as in section 3, only with $A = 2$ or $A = 1$, see appendix B. This is because computing the contraction is much more demanding compared to the fluid cell averaging case, hence we used smaller $A$ to speed up simulations. Note that we did not simulate the significantly more complicated full time evolution, see appendix B.

## 5   Diffusive generalized hydrodynamics

The above results show that there is no diffusive scale $1/\ell$ correction to generalized hydrodynamics in hard rods (on the level of individual samples). However, one typically assumes that

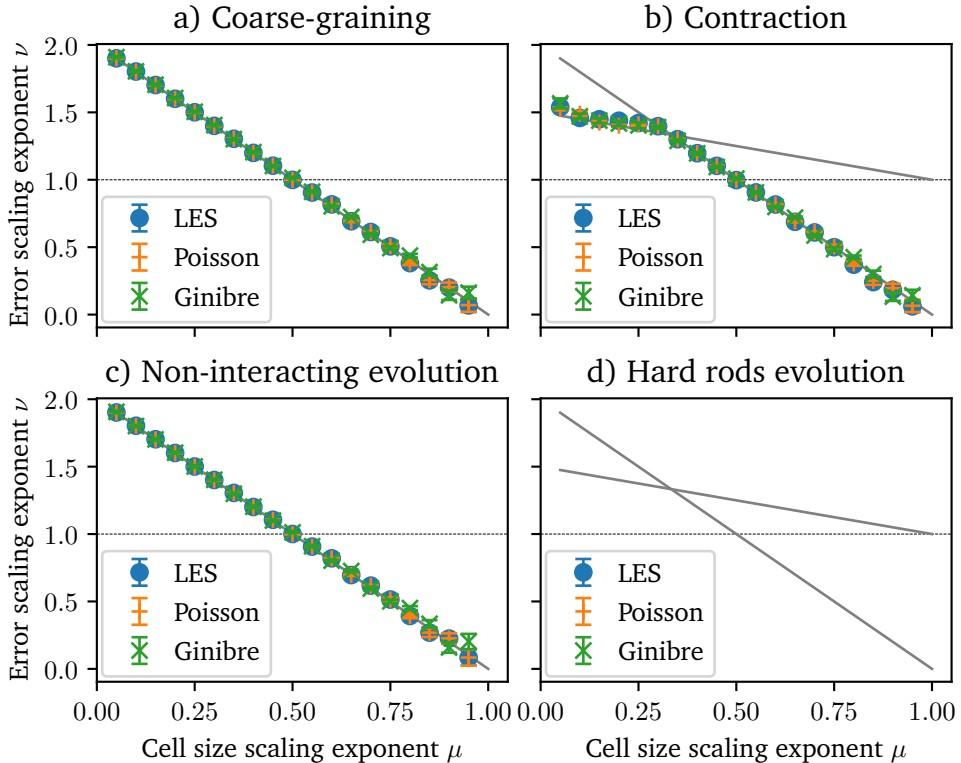

Figure 3: Numerically extracted scaling of the error $\xi \sim \ell^{-\nu}$ as function of coarse-graining scale $\Delta x, \Delta p \sim \ell^{\mu-1}$ for smoothing coarse-graining. We dististinguish the four approximations discussed in the text: a) coarse-graining of the initial configuration (section 4.1), b) contracting map (section 4.2), c) non-interacting time evolution (section 4.3). We were not able to compute numerical data for d) the full hard rods time evolution (section 4.4) since it was too numerically demanding (see appendix B). The points are the numerical results for the three ensembles discussed in section 3.5. The errorbars represent the fit error on $\nu$ (note that the error for the Ginibre states are larger, since they had to be obtained on smaller system sizes). The numerical results agree well with the analytical predictions obtained in this section. The dashed line at $\nu = 1$ is diffusive scaling. If $\nu > 1$, Euler hydrodynamics is accurate also on the diffusive scale. Further details of the numerical simulations are explained in appendix B.

the initial configuration is chosen from some ensemble of initial states, e.g. local equilibrium states like (2). The quantity of interest is then the quasi-particle density $\langle \rho(t, x, p) \rangle$ averaged over this initial ensemble. We will now show that the evolution equation of $\langle \rho(t, x, p) \rangle$ is not given by the Euler GHD equation (8), but indeed obtains a $1/\ell$ correction (which, however, is different from a Navier-Stokes like diffusion term).

The appearance of such a term can easily be seen as follows: Since Euler GHD is accurate beyond $1/\ell$, we can forget about the microscopic hard rods model and study Euler GHD with random initial data. In hard rods the solution to the GHD equation is known explicitly and given by (12). Thus, all we need to do is to average (12) over the initial data. To obtain a systematic expansion, we will use that fluctuations are small along the following strategy.

Imagine the following toy example: given a deterministic map $y = f(x)$, we want to compute the average of $y$ over a probability distribution of $x$. If this probability distribution is peaked at $\langle x \rangle$ with a small fluctuations $\delta x = x - \langle x \rangle \sim \varepsilon \ll 1$, we can compute this via Taylor

expansion

$$\langle y \rangle = \langle f(x) \rangle = \langle f(\langle x \rangle + \delta x) \rangle = f(\langle x \rangle) + f'(\langle x \rangle) \underbrace{\langle \delta x \rangle}_{=0} + \tfrac{1}{2} f''(\langle x \rangle) \langle \delta x^2 \rangle + \mathcal{O}(\varepsilon^3). \quad (128)$$

In our case, $x$ stands for $\rho(x,p)$ and $f$ should be seen as the GHD equation solution map (12). The fluctuations of $\delta\rho(x,p) = \rho(x,p) - \langle\rho(x,p)\rangle$ are typically of order $\varepsilon = 1/\sqrt{\ell}$, hence the correction term on the RHS will be of diffusive order $1/\ell$. This is what we will call diffusive GHD.

## 5.1 Time evolved two-point correlation function in hard rods

From (128) it is clear that the diffusive scale correction is determined by $\langle\delta\rho(x,p)\delta\rho(y,q)\rangle$, i.e. the two point correlation functions. Here we briefly summarize the current understanding of the two-point function as derived from GHD [22, 52]. As mentioned above a natural initial state for the GHD is a local equilibrium state. In these states the two-point function $\langle\delta\rho(x,p)\delta\rho(y,q)\rangle$ (also denoted $\langle\rho(x,p)\rho(y,q)\rangle^c$) is uncorrelated in space and is locally given by the local GGE correlations:

$$\langle\delta\rho(x,p)\delta\rho(y,q)\rangle = \tfrac{1}{\ell}\delta(x-y)C_{\text{GGE}}(x,p,q) + \mathcal{O}(1/\ell^2), \quad (129)$$

where

$$C_{\text{GGE}}(x,p,q) = \rho(x,p)\delta(p-q) + \rho(x,p)\rho(x,q)\left(-2a + a^2\int dq'\,\rho(x,q')\right) \quad (130)$$

are the GGE correlation functions. During time evolution the form of this singular part is preserved, however, additional long range correlations emerge. The time evolved two-point correlation function is thus given by

$$C(x,p,y,q) = \ell\,\langle\delta\rho(x,p)\delta\rho(y,q)\rangle = \delta(x-y)C_{\text{GGE}}(x,p,q) + C_{\text{LR}}(x,p,y,q) + \mathcal{O}(1/\ell). \quad (131)$$

One can find explicit formulas for the long range part [22, 52], however, they will not be important here. To derive diffusion only the local behaviour at $x \approx y$ will be important. It turns out that the long range correlations (which are otherwise continuous) have a jump at $x = y$:

$$C_{\text{LR}}(x,p,y,q)\big|_{y\approx x} = \tfrac{a}{2}1^{\text{dr}}(x)\big[\partial_x\rho(x,p)\rho(x,q) - $$
$$ - \rho(x,p)\partial_x\rho(x,q)\big]\text{sgn}(y-x) + (\text{continuous}). \quad (132)$$

Here (continuous) stands for a continuous part and $1^{\text{dr}}(x) = 1 - a\int dq\,\rho(x,q)$. We will see that the contribution from the jump and from the $\delta(x-y)$ term will cancel, and thus only the continuous part will affect the evolution.

## 5.2 The diffusive scale correction

In order to apply the above strategy outlined in (128) we start from (12) and proceed as follows:

$$\Upsilon(t) = \left\langle \int dx\,dp\,\rho(x,p)\Upsilon(X(t,x,p),p) \right\rangle \quad (133)$$

$$= \int dx\,dp\,\langle\rho(x,p)\rangle\,\Upsilon(\langle X(t,x,p)\rangle,p) + \int dx\,dp\,\partial_x\Upsilon(\langle X(t,x,p)\rangle,p)\langle\delta\rho(x,p)\delta X(t,x,p)\rangle$$

$$+ \tfrac{1}{2}\int dx\,dp\,\langle\rho(x,p)\rangle\,\partial_x^2\Upsilon(\langle X(t,x,p)\rangle,p)\langle\delta X(t,x,p)^2\rangle + \mathcal{O}(1/\ell^2). \quad (134)$$

Here we defined $\delta X(t,x,p) = X(t,x,p) - \langle X(t,x,p)\rangle$. Since $X(t,x,p)$ is also a non-linear function of $\rho(x,p)$, $\langle X(t,x,p)\rangle$ will also obtains a $1/\ell$ correction. We will compute its value later, but for now let us denote it by $\Delta X(t,x,p)$, i.e.

$$\langle X(t,x,p)\rangle = X_{\mathrm{E}}(t,x,p) + \tfrac{1}{\ell}\Delta X(t,x,p) + \mathcal{O}\big(1/\ell^2\big). \tag{135}$$

Here $X_{\mathrm{E}}(t,x,p)$ is (10), but with $\rho$ replaced by $\langle\rho\rangle$. Inserting this into (134) gives

$$
\begin{aligned}
\Upsilon(t) = &\int \mathrm{d}x\,\mathrm{d}p\,\langle\rho(x,p)\rangle\,\Upsilon(X_{\mathrm{E}}(t,x,p),p) \\
&+ \tfrac{1}{\ell}\int \mathrm{d}x\,\mathrm{d}p\,\partial_x\Upsilon(X_{\mathrm{E}}(t,x,p),p)[\langle\rho(x,p)\rangle\,\Delta X(t,x,p) + \ell\,\langle\delta\rho(x,p)\delta X(t,x,p)\rangle] \\
&+ \tfrac{1}{2\ell}\int \mathrm{d}x\,\mathrm{d}p\,\langle\rho(x,p)\rangle\,\partial_x^2\Upsilon(X_{\mathrm{E}}(t,x,p),p)\ell\,\big\langle\delta X(t,x,p)^2\big\rangle + \mathcal{O}\big(1/\ell^2\big).
\end{aligned}
\tag{136}
$$

We will denote $B(t,x,p) = \langle\rho(x,p)\rangle\,\Delta X(t,x,p) + \ell\,\langle\delta\rho(x,p)\delta X(t,x,p)\rangle$.

As we are interested in finding the evolution equation for $\langle\rho_t(x,p)\rangle$, we need to study the behavior of this as $t \to 0^+$. We show in appendix A that

$$\lim_{t\to 0^+} X_{\mathrm{E}}(t,x,p) = x \tag{137}$$

$$\lim_{t\to 0^+} B(t,x,p) = 0 \tag{138}$$

$$\lim_{t\to 0^+} \ell\,\big\langle\delta X(t,x,p)^2\big\rangle = 0, \tag{139}$$

hence as $t \to 0^+$ we have

$$\lim_{t\to 0^+}\Upsilon(t) = \int \mathrm{d}x\,\mathrm{d}p\,\rho(x,p)\Upsilon(x,p) \tag{140}$$

as expected. Next, let us take a time derivative of (136) at $t = 0^+$. We find

$$
\begin{aligned}
\frac{\mathrm{d}}{\mathrm{d}t}\Upsilon(t)\bigg|_{t=0^+} = &\int \mathrm{d}x\,\mathrm{d}p\,\langle\rho(x,p)\rangle\,\partial_x\Upsilon(x,p)v^{\mathrm{eff}}(x,p) \\
&+ \tfrac{1}{\ell}\int \mathrm{d}x\,\mathrm{d}p\,\partial_x\Upsilon(x,p)\frac{\mathrm{d}}{\mathrm{d}t}B(t,x,p)\bigg|_{t=0^+} \\
&+ \tfrac{1}{2\ell}\int \mathrm{d}x\,\mathrm{d}p\,\langle\rho(x,p)\rangle\,\partial_x^2\Upsilon(x,p)\frac{\mathrm{d}}{\mathrm{d}t}\big[\ell\,\big\langle\delta X(t,x,p)^2\big\rangle\big]\bigg|_{t=0^+} + \mathcal{O}\big(1/\ell^2\big).
\end{aligned}
\tag{141}
$$

Here we used that $\frac{\mathrm{d}}{\mathrm{d}t}X_{\mathrm{E}}(t,x,p) = v^{\mathrm{eff}}(x,p)$. Since (141) has to hold for any observable $\Upsilon$, this implies

$$
\begin{aligned}
\partial_t\langle\rho_t(x,p)\rangle\bigg|_{t=0^+} = &-\partial_x\big(v^{\mathrm{eff}}(x,p)\langle\rho(x,p)\rangle\big) - \tfrac{1}{\ell}\partial_x\bigg(\frac{\mathrm{d}}{\mathrm{d}t}B(t,x,p)\bigg|_{t=0^+}\bigg) \\
&+ \tfrac{1}{2\ell}\partial_x^2\bigg[\langle\rho(x,p)\rangle\frac{\mathrm{d}}{\mathrm{d}t}\big[\ell\,\big\langle\delta X(t,x,p)^2\big\rangle\big]\bigg|_{t=0^+}\bigg] + \mathcal{O}\big(1/\ell^2\big).
\end{aligned}
\tag{142}
$$

This is the GHD equation including the $1/\ell$ correction:

$$\partial_t\langle\rho_t(x,p)\rangle\bigg|_{t=0^+} + \partial_x\big(v^{\mathrm{eff}}(x,p)\langle\rho(x,p)\rangle\big) + \tfrac{1}{\ell}\partial_x(\Delta j(x,p)) = 0, \tag{143}$$

where

$$\Delta j(x,p) = \frac{\mathrm{d}}{\mathrm{d}t} B(t,x,p)\Big|_{t=0^+} - \tfrac{1}{2}\partial_x\left[ \langle \rho(x,p)\rangle \frac{\mathrm{d}}{\mathrm{d}t}\big[\ell \langle \delta X(t,x,p)^2\rangle\big]\Big|_{t=0^+}\right] \tag{144}$$

is the correction to the current.

We still need to relate $B(t,x,p)$ and $\ell\langle\delta X(t,x,p)^2\rangle$ to $\langle\delta\rho(x,p)\delta\rho(y,q)\rangle$. Let us start by computing

$$\langle X(t,x,p)\rangle = \left\langle \hat{X}(x) + pt + a\int \mathrm{d}y\,\mathrm{d}q\,\rho(y,q)\theta(\hat{X}(x)+pt-\hat{X}(y)-qt) - \tfrac{a}{2\ell}\right\rangle \tag{145}$$

$$= \langle\hat{X}(x)\rangle + pt + a\int \mathrm{d}y\,\mathrm{d}q\,\langle\rho(y,q)\rangle\,\theta(\langle\hat{X}(x)\rangle+pt-\langle\hat{X}(y)\rangle-qt) - \tfrac{a}{2\ell}$$

$$+ a\int \mathrm{d}y\,\mathrm{d}q\,\delta(\langle\hat{X}(x)\rangle+pt-\langle\hat{X}(y)\rangle-qt)\langle\delta\rho(y,q)(\delta\hat{X}(x)-\delta\hat{X}(y))\rangle$$

$$+ \tfrac{a}{2}\int \mathrm{d}y\,\mathrm{d}q\,\langle\rho(y,q)\rangle\,\delta'(\langle\hat{X}(x)\rangle+pt-\langle\hat{X}(y)\rangle-qt)\langle(\delta\hat{X}(x)-\delta\hat{X}(y))^2\rangle + \dots \tag{146}$$

Here we denoted by $\delta\hat{X}(x) = \hat{X}(x)-\langle\hat{X}(x)\rangle = -a\int \mathrm{d}y\,\delta\bar{\rho}(y)\theta(x-y)$, with $\bar{\rho}(y) = \int \mathrm{d}q\,\rho(y,q)$. Also observe $1^{\mathrm{dr}}(x) = 1-a\bar{\rho}(x) = \partial_x\hat{X}(x)$ and $X(\hat{x})$ as to inverse function to $\hat{X}(x)$. Comparing (146) with (135) we can read off

$$\Delta X(t,x,p) = a\int \mathrm{d}q\,\frac{1}{1^{\mathrm{dr}}(y)}\ell\left\langle\delta\rho(y,q)(\delta\hat{X}(x)-\delta\hat{X}(y))\right\rangle\Big|_{y=X(\hat{X}(x)+(p-q)t)}$$

$$+ \tfrac{a}{2}\int \mathrm{d}q\,\frac{1}{1^{\mathrm{dr}}(y)}\partial_y\left(\frac{\rho(y,q)}{1^{\mathrm{dr}}(y)}\ell\left\langle(\delta\hat{X}(x)-\delta\hat{X}(y))^2\right\rangle\right)\Big|_{y=X(\hat{X}(x)+(p-q)t)} \tag{147}$$

$$= a\int \mathrm{d}q\,\mathrm{d}q'\,\frac{1}{1^{\mathrm{dr}}(y)}\Big(\delta(q-q') + a\frac{\rho(y,q)}{1^{\mathrm{dr}}(y)}\Big)\ell\left\langle\delta\rho(y,q')(\delta\hat{X}(x)-\delta\hat{X}(y))\right\rangle\Big|_{y=X(\hat{X}(x)+(p-q)t)}$$

$$+ \tfrac{a}{2}\int \mathrm{d}q\,\frac{\partial_y\frac{\rho(y,q)}{1^{\mathrm{dr}}(y)}}{1^{\mathrm{dr}}(y)}\ell\left\langle(\delta\hat{X}(x)-\delta\hat{X}(y))^2\right\rangle\Big|_{y=X(\hat{X}(x)+(p-q)t)}. \tag{148}$$

Here and from now on, for lightness of notation, we decided to drop the $\langle\dots\rangle$ around one-point functions. In terms of the initial fluctuations the expectation values are given by

$$\langle\delta\hat{X}(x)\delta\hat{X}(y)\rangle = a^2\int_{-\infty}^{x}\mathrm{d}z_1\int_{-\infty}^{y}\mathrm{d}z_2\,\langle\delta\bar{\rho}(z_1)\delta\bar{\rho}(z_2)\rangle \tag{149}$$

$$\langle\delta\rho(y,q)\delta\hat{X}(x)\rangle = -a\int_{-\infty}^{x}\mathrm{d}z_1\,\langle\delta\rho(y,q)\delta\bar{\rho}(z_1)\rangle \tag{150}$$

$$\langle\delta\rho(y,q)\delta\hat{X}(y)\rangle = -a\int_{-\infty}^{y}\mathrm{d}z_1\,\langle\delta\rho(y,q)\delta\bar{\rho}(z_1)\rangle. \tag{151}$$

Furthermore, we can compute $\delta X(t,x,p)$ as

$$\delta X(t,x,p) = \delta\hat{X}(x) + a\int dy\,dq\,\delta\rho(y,q)\theta(\hat{X}(x)+pt-\hat{X}(y)-qt)$$

$$+ a\int dy\,dq\,\rho(y,q)\delta(\hat{X}(x)+pt-\hat{X}(y)-qt)\big(\delta\hat{X}(x)-\delta\hat{X}(y)\big) \tag{152}$$

$$= \delta\hat{X}(x) + a\int dy\,dq\,\theta(\hat{X}(x)+pt-\hat{X}(y)-qt)\delta\rho(y,q)$$

$$+ a\int dq\,\frac{\rho(y,q)}{1^{\mathrm{dr}}(y)}\big(\delta\hat{X}(x)-\delta\hat{X}(y)\big)\bigg|_{y=X(\hat{X}(x)+(p-q)t)}. \tag{153}$$

Hence,

$$\langle\delta\rho(x,p)\delta X(t,x,p)\rangle = \langle\delta\rho(x,p)\delta\hat{X}(x)\rangle$$

$$+ a\int dy\,dq\,\theta(\hat{X}(x)+pt-\hat{X}(y)-qt)\langle\delta\rho(x,p)\delta\rho(y,q)\rangle$$

$$+ a\int dq\,\frac{\rho(y,q)}{1^{\mathrm{dr}}(y)}\big\langle\delta\rho(x,p)\big(\delta\hat{X}(x)-\delta\hat{X}(y)\big)\big\rangle\bigg|_{y=X(\hat{X}(x)+(p-q)t)} \tag{154}$$

and

$$\big\langle\delta X(t,x,p)^2\big\rangle = \big\langle\delta\hat{X}(x)^2\big\rangle$$

$$+ a^2\int dy\,dq\,dy'\,dq'\,\theta(\hat{X}(x)+pt-\hat{X}(y)-qt)\theta(\hat{X}(x)+pt-\hat{X}(y')-q't)$$

$$\times\big\langle\delta\rho(y,q)\delta\rho(y',q')\big\rangle$$

$$+ a^2\int dq\,dq'\,\frac{\rho(y,q)}{1^{\mathrm{dr}}(y)}\frac{\rho(y',q')}{1^{\mathrm{dr}}(y')}$$

$$\times\big\langle\big(\delta\hat{X}(x)-\delta\hat{X}(y)\big)\big(\delta\hat{X}(x)-\delta\hat{X}(y')\big)\big\rangle\bigg|_{y=X(\hat{X}(x)+(p-q)t),y'=X(\hat{X}(x)+(p-q')t)}$$

$$+ 2a\int dy\,dq\,\theta(\hat{X}(x)+pt-\hat{X}(y)-qt)\big\langle\delta\hat{X}(x)\delta\rho(y,q)\big\rangle$$

$$+ 2a\int dq\,\frac{\rho(y,q)}{1^{\mathrm{dr}}(y)}\big\langle\delta\hat{X}(x)\big(\delta\hat{X}(x)-\delta\hat{X}(y)\big)\big\rangle\bigg|_{y=X(\hat{X}(x)+(p-q)t)}$$

$$+ 2a^2\int dy\,dq\,dq'\,\theta(\hat{X}(x)+pt-\hat{X}(y)-qt)\frac{\rho(y',q')}{1^{\mathrm{dr}}(y')}$$

$$\times\big\langle\delta\rho(y,q)\big(\delta\hat{X}(x)-\delta\hat{X}(y')\big)\big\rangle\bigg|_{y'=X(\hat{X}(x)+(p-q')t)}. \tag{155}$$

At this point we have all necessary ingredients, i.e. $\Delta X(t,x,p)$, $\langle\delta\rho(x,p)\delta X(t,x,p)\rangle$ and $\big\langle\delta X(t,x,p)^2\big\rangle$, to compute (144) in terms of the initial fluctuations. This is done in appendix A. Since the formulas are linear in the correlations we can compute the effect of each contribution individually.

The contribution from the singular part of the correlations is given by

$$\Delta j(x,p)\overset{\mathrm{singular}}{=}-\tfrac{1}{2}\hat{\mathbf{D}}\partial_x\rho(x,p) \tag{156}$$

and the contribution by the long range part by

$$\Delta j(x,p) \overset{\text{long range}}{=} \frac{a}{1^{\text{dr}}(x)} \int dq \, dp' \, dq' (p-q) \Big( \delta(p-p') + a \frac{\rho(x,p)}{1^{\text{dr}}(x)} \Big)$$

$$\times \Big( \delta(q-q') + a \frac{\rho(x,q)}{1^{\text{dr}}(x)} \Big) C_{\text{LR}}(x - v^{\text{eff}}(x,p)0^+, p', x - v^{\text{eff}}(x,q)0^+, q'). \quad (157)$$

This is identical to the diffusive scale correction obtained in [22]. As noticed in [22], the contribution from the singular and the jump of the long range correlations exactly cancel. Thus the only contribution comes from the regular part of the long range correlations

$$C_{\text{LR,sym}}(x,p,x,q) = \frac{1}{2} \Big( C_{\text{LR}}(x+0^+,p,x,q) + C_{\text{LR}}(x+0^-,p,x,q) \Big), \quad (158)$$

i.e.:

$$\partial_t \rho(x,p) + \partial_x (v^{\text{eff}}(x,p)\rho(x,p)) = -\frac{1}{\ell} \partial_x \Big[ \frac{a}{1^{\text{dr}}(x)} \int dq \, dp' \, dq' (p-q) \quad (159)$$

$$\times \Big( \delta(p-p') + a \frac{\rho(x,p)}{1^{\text{dr}}(x)} \Big) \Big( \delta(q-q') + a \frac{\rho(x,q)}{1^{\text{dr}}(x)} \Big) C_{\text{LR,sym}}(x,p',x,q') \Big]. \quad (160)$$

This is the diffusive scale correction to GHD in hard rods and it is discussed in more detail in [18, 22]. Note that it is not given by a Navier-Stokes like diffusion term. In particular, (160) combined with an equation for the correlation functions is invariant under time reversal symmetry. This is in stark contrast to the Navier-Stokes like diffusion, which increases entropy (and thus cannot be time-reversible). Also note that the diffusive scale correction purely is determined by the correlations.

**Remark 9** *Even though* (160) *is called the diffusive GHD equation, its RHS is not at all of diffusive type. This is because there is no intrinsic diffusion. All correction terms are due to transport of initial fluctuations, i.e. "diffusion from convection". Since this transport is time-reversible it cannot give rise to an diffusive expression.*

At this point, we should clarify the practical meaning of this result: imagine an experiment where the evolution of one specific configuration of hard rods is observed. Then this should be described by the Euler GHD equation (8) without diffusive correction, since intrinsic diffusion is absent. If on the other hand, one is able to only measure observables averaged over many initial states, then one should use (160) instead. This kind of thinking is even more important for quantum systems: a quantum system can either be prepared in a pure state (for instance on a quantum computer) or in a mixed state (consider a cold atom experiment). In the latter case the quantum state of the particles is a mixed state, even in theory.

# 6 Entropy increase from the perspective of hydrodynamics without averaging

Due to the absence of an intrinsic diffusive correction to the GHD of hard rods, entropy will remain constant for all times[6]. In particular, there seems to be no thermalization towards a GGE. Before the realization that the diffusive scale correction to GHD is not given by an entropy increasing Navier-Stokes like equation, it was believed that this diffusive equation would lead to thermalization. Since the Navier-Stokes term is suppressed by $1/\ell$, the expected time scale of thermalization was $T \sim \ell$ in macroscopic units (or $T \sim \ell^2$ in microscopic units).

---

[6]The Euler scale hydrodynamic equations conserve entropy [2].

However, from the perspective of hydrodynamics without averaging it is impossible to reach such long times. Already for non-interacting particles, we can see from (73) and (117) if $t \gg 1$, the error is controlled by $\Delta p t$. Therefore, the hydrodynamic approximation certainly breaks down when $t \sim 1/\Delta p \ll \ell$. This makes sense since at this time, particles that initially were in the same fluid cell are macroscopically apart.

On this time scale relaxation to a GGE state will occur, simply because the coarse-graining is not able to capture all the details of the initial configuration anymore. In the following we would like to explain this on the simple example of non-interacting particles, i.e. $a \to 0$. Due to the mapping to non-interacting particles, the result should in spirit carry over to hard rods (and any integrable model).

**Remark 10** *The ideas and derivation presented here are very close to another work published in 2022 [53] (in particular the derivation via fluid cell coarse-graining). This section should therefore not be seen as original work, but rather a reinterpretation of their result in the context of this paper.*

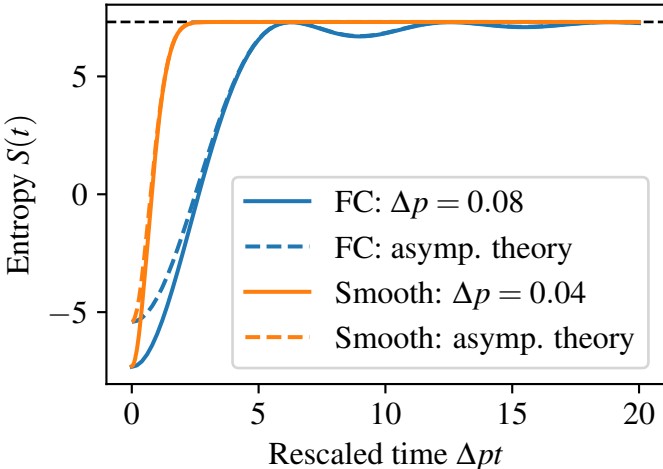

Figure 4: Evolution of the coarse-grained (classical) entropy starting from $\rho^0(x, p) = (10 - 9 \sin(x)) e^{-p^2/2} / \sqrt{2\pi}$ (the system is a periodic box of size $2\pi$) at long times $t = \mathcal{O}(1/\Delta p) \gg 1$ much beyond Euler scale. For fluid cell coarse-graining we used $\Delta p = 0.08$ and $\Delta x = 2\pi/100$ and for smooth coarse-graining we used $\Delta p = 0.04$ and $\Delta x = 2\pi/200$. In both cases we observe thermalization, however, the approach depends on the coarse-graining scheme. For smoothing it thermalizes quickly and monotonously, while for fluid cell coarse-graining it thermalizes slowly and in an oscillatory fashion. The transition is well described by the asymptotic expressions (dashed lines) derived in the text.

## 6.1 Fluid cell coarse-graining

To ensure a finite density even for long times we study a periodic box of size $2\pi$[7]. We will ignore the effect of the initial coarse-graining and start directly from a smooth initial quasi-particle density $\rho(x, p)$. The averaged density at time $t$ in cell $C_{\alpha,\beta}$ is then

$$\rho_{\alpha,\beta}(t) = \int_{-1/2}^{1/2} dy \, dq \, \rho^0(x_\alpha - p_\beta t + y \Delta x - q \Delta p t, p_\beta + q \Delta p) =: E_{FC}(t, x_\alpha - p_\beta t, p_\beta).$$

(161)

---

[7]If we study the problem on the real line, particles will become infinitely separated for $t \gg 1$.

As $\Delta pt \to \infty$ we have

$$E_{\text{FC}}(t,x,p) = \int_{-1/2}^{1/2} \mathrm{d}y \, \mathrm{d}q \, \rho^0(x + y\Delta x - q\Delta pt, p + q\Delta p) \approx \int_{-1/2}^{1/2} \mathrm{d}q \, \rho^0(x - q\Delta pt, p). \tag{162}$$

For convenience, we also send the subleading $\Delta x \sim \Delta p \to 0$. Now expand

$$\rho(x,p) = \sum_{k=-\infty}^{\infty} \tilde{\rho}_k(p) e^{ikx} \tag{163}$$

in Fourier components

$$E_{\text{FC}}(t,x,p) \approx \sum_k \int_{-1/2}^{1/2} \mathrm{d}q \, \tilde{\rho}_k(p) e^{ikx - ikq\Delta pt} = \tilde{\rho}_0(p) + \sum_{k \neq 0} \tilde{\rho}_k(p) \frac{\sin(k\Delta pt/2)}{k\Delta pt/2} e^{ikx}. \tag{164}$$

For $\Delta pt \to \infty$ the second term vanishes and (as expected) the density converges to the constant value of the GGE $\tilde{\rho}_0(p) = \int_0^{2\pi} \frac{\mathrm{d}x}{2\pi} \rho(x,p)$. We can study the approach to equilibrium using the total coarse-grained entropy

$$S_{\text{FC}}(t) = -\Delta x \Delta p \sum_{\alpha,\beta} \gamma(\rho_{\alpha,\beta}(t)) = -\Delta x \Delta p \sum_{\alpha,\beta} \gamma(E_{\text{FC}}(t, x_\alpha + p_\beta t, p_\beta)) \tag{165}$$

$$\to -\int \mathrm{d}x \, \mathrm{d}p \, \gamma(E_{\text{FC}}(t,x,p)) = -\int \mathrm{d}x \, \mathrm{d}p \, \gamma\left( \tilde{\rho}_0(p) + \sum_{k \neq 0} \tilde{\rho}_k(p) \frac{\sin(k\Delta pt/2)}{k\Delta pt/2} e^{ikx} \right) \tag{166}$$

$$= S(\infty) - \frac{2\pi}{2} \int \mathrm{d}p \, \gamma''(\tilde{\rho}_0(p)) \sum_{k \neq 0} |\tilde{\rho}_k(p)|^2 \left( \frac{\sin(k\Delta pt/2)}{k\Delta pt/2} \right)^2 + \mathcal{O}\big(1/(\Delta pt)^3\big), \tag{167}$$

where $\gamma(\rho)$ depends on the particle statistics. For classical particles it is given by $\gamma(\rho) = -\rho \log \rho + \rho$. The final value $S(\infty) = 2\pi \int \mathrm{d}p \, \gamma(\tilde{\rho}_0(p))$ is the GGE entropy. Note that the entropy approaches the equilibrium non-monotonously(!) on the time-scale $T_{\text{th}} \sim 1/\Delta p$, as expected (see fig. 4).

## 6.2   Smoothing

Under smoothing we find that

$$\rho_{\text{Smooth}}(t,x,p) = \int \mathrm{d}y \, \mathrm{d}q \, \eta(y,q) \rho(x - pt + y\Delta x - q\Delta pt, p + q\Delta p) \tag{168}$$

$$=: E_{\text{Smooth}}(t, x - pt, p). \tag{169}$$

As $\Delta pt \to \infty$ we have

$$E_{\text{Smooth}}(t,x,p) \approx \int \mathrm{d}q \, \eta_{\text{p}}(q) \rho(x - q\Delta pt, p), \tag{170}$$

where we defined $\eta_{\text{p}}(q) = \int \mathrm{d}y \, \eta(y,q)$. Expanding again $\rho(x,p) = \sum_{k=-\infty}^{\infty} \tilde{\rho}_k(p) e^{ikx}$ in Fourier components we get

$$E_{\text{Smooth}}(t,x,p) \approx \sum_{k=-\infty}^{\infty} \tilde{\rho}_k(p) e^{ikx} \tilde{\eta}_{\text{p}}(k\Delta pt) \tag{171}$$

$$= \tilde{\rho}_0(p) + \sum_{k \neq 0} \tilde{\rho}_k(p) e^{ikx} \tilde{\eta}_{\text{p}}(k\Delta pt), \tag{172}$$

where $\tilde{\eta}_{\mathrm{p}}(z) = \int \mathrm{d}q\, \eta_{\mathrm{p}}(q) e^{-iqz}$. Inserting this into the coarse-grained entropy

$$S_{\mathrm{Smooth}}(t) = -\int \mathrm{d}x\, \mathrm{d}p\, \gamma(\rho_{\mathrm{Smooth}}(t,x,p)) = -\int \mathrm{d}x\, \mathrm{d}p\, \gamma(E_{\mathrm{Smooth}}(t,x,p)) \qquad (173)$$

$$= S(\infty) - \frac{2\pi}{2} \int \mathrm{d}p\, \gamma''(\tilde{\rho}_0(p)) \sum_{k \neq 0} |\tilde{\rho}_k(p)|^2 \left|\tilde{\eta}_{\mathrm{p}}(k\Delta p t)\right|^2 + \ldots \qquad (174)$$

$$= S(\infty) - 2\pi \int \mathrm{d}p\, \gamma''(\tilde{\rho}_0(p)) |\tilde{\rho}_1(p)|^2 \left|\tilde{\eta}_{\mathrm{p}}(\Delta p t)\right|^2 + \ldots \qquad (175)$$

Note that here higher $k$ Fourier components decay significantly faster than the $k = \pm 1$ mode. This is different from (167), where all Fourier modes decay equally fast. Furthermore, the approach to the final value is much faster for a smooth $\eta_{\mathrm{p}}(p)$ than for the fluid cell coarse-graining. We plot this entropy in fig. 4 with coarse-graining function $\eta_{\mathrm{p}}(p) = \frac{1}{\sqrt{2\pi}} e^{-\frac{1}{2}p^2}$. Since the Fourier transform of this function is also a Gaussian, the resulting entropy is monotonically increasing in time. However, choosing a different $\eta_{\mathrm{p}}(p)$ can lead to an oscillating approach to the final value.

From these derivations it is clear that the approach to the thermal equilibrium depends on the coarse-graining scheme. This is natural if the coarse-graining is interpreted as the measurement imprecision of a measurement device: from the viewpoint of an experimentalist the system seems thermal. However, from an abstract viewpoint this is very unsatisfactory: thermalization should somehow be a universal phenomena, and not be due to insufficient measurement devices.

We can imagine that this is an artifact of the non-existence of gradient catastrophes in integrable models: in non-integrable models, gradient catastrophes like shocks create entropy, thereby leading to much quicker thermalization already at the Euler scale. Therefore, the spurious coarse-graining dependent entropy increase will likely not be important in non-integrable models. However, further studies would be necessary.

## 7 Discussion

We have presented a new paradigm to study hydrodynamic approximations in deterministic systems. We dubbed it "hydrodynamics without averaging" and carried it out in hard rods, an exactly solvable model. The idea of the paradigm is to abandon initial state randomness and to consider a fixed deterministic initial configuration. This way, there is no explicit randomness, neither in the initial state, nor in the evolution. In order to find a meaningful initial density profile for the hydrodynamic evolution one has to coarse-grain the initial state. At least on an intuitive level, this procedure is well established in hydrodynamics. However, it was not carried out in practice. Instead, one usually considers random initial states, such as (2). They have the advantage that many mathematical details are much simpler: for instance, no coarse-graining is required and also the averaged densities are smooth. Also, one can clearly define the Euler scaling limit by taking the spatial variation scale $\ell \to \infty$ and scaling time and particle number appropriately. This means that one can clearly identify the limiting values as well as correction terms.

Compared to that, "hydrodynamics without averaging" is significantly more ambiguous: First, there are many different coarse-graining strategies (here we discussed two: fluid cell coarse-graining and smoothing) leading to potentially different results. Second, after coarse-graining the density profile becomes rough (on the macroscopic scale), meaning that it is important to understand how to evolve such rough solutions (i.e. for instance by means of

weak solutions). And third, it is not straight-forward to estimate the asymptotic scaling of the error, because configurations for two different $\ell$ can be very different.

On the explicit example of hard rods we discussed these problems: we quantify the error by comparing the value of a (smooth) observable in the hydrodynamic theory to the one in the microscopic theory. We show that the scaling of the error can still be meaningfully predicted by making assumptions of local self-averaging. This means that our derivations will apply only to locally sufficiently generic states. That this is justified is demonstrated by comparing to numerical simulations: these numerical simulations agree very well, not just for physical initial states, but also for unphysical initial states. This shows that locally generic does not require local equilibrium. In fact, the local equilibrium assumption of hydrodynamics is not required to be able to apply hydrodynamics (at least in this model).

The scalings of the error show a "phase-transition". For large coarse-graining scales $\Delta x$ the error is dominated by a systematic error scaling as $\mathcal{O}(\Delta x^2)$. For smaller $\Delta x$ the error is instead dominated by a statistical error (depending on the coarse-graining procedure). Even without time evolution this highlights an important limitation of hydrodynamics solely due to coarse-graining: both the coarse-graining procedures discussed are only able to reproduce the value of an observable up to an error at least $\mathcal{O}(\Delta x^2)$. This has strong limitations on the validity of higher-order corrections: As $\Delta x \gg 1/\ell$, this means that any hydrodynamic theory based on these coarse-graining schemes will always have an error larger than $\mathcal{O}(1/\ell^2)$. Therefore, a hydrodynamic theory can in principle capture diffusive corrections $\mathcal{O}(1/\ell)$, but no higher order corrections such as a dispersive one $\mathcal{O}(1/\ell^2)$. To be able to understand such corrections one would need to know more information about the initial state (for instance the average location of particles inside fluid cells). This strongly suggest that higher order corrections to hydrodynamics are not just equations of the densities, but coupled to additional equations describing these additional degrees of freedom, contradicting the current intuitive expectation, see e.g. [54, Eq. (103)] and [55]. There are, however, potential alternatives. For instance, it might be that there are coarse-graining schemes that are more accurate or it might be required to use a different definition of the densities: densities are always only defined up to a total derivative, meaning that there might be a more accurate density. It would be interesting to search for better schemes or to show that no better schemes exist.

Gaining insights into questions like the above is a major advantage of "hydrodynamics without averaging". In the more traditional approach, where one averages over the initial configuration, all order corrections to the expectation value at later times are in principle defined. However, these higher order corrections might then fail to describe the actual physics because these higher orders cannot self-average. In this regard there is also another major advantage: since "hydrodynamics without averaging" establishes the validity of hydrodynamics for each deterministic configuration, randomness in the initial state can be taken into account later, simply by averaging over the hydrodynamic evolution. This establishes also the validity of BMFT (ballistic macroscopic fluctuation theory), which had been established previously as a way to compute hydrodynamic correlation functions. Interestingly, we found that for sufficiently small coarse-graining Euler hydrodynamics is more accurate than a diffusive correction $\mathcal{O}(1/\ell)$. This means that there is no (intrinsic) diffusive correction to hydrodynamics in hard rods. However, if one averages over the initial configurations, then the expectation value at later times has a $1/\ell$ correction, stemming from the fact that the initial randomness is transported non-linearly along the hydrodynamic modes. We derive this correction and find that it coincides with the recently establish diffusive scale correction in hard rods [18, 22]. This diffusive scale correction was surprising, because it does not have the form of the usual diffusion term expected in hydrodynamics. From the perspective of "hydrodynamics without averaging" the appearance of such an unconventional correction on the diffusive scale is clear: it is only an artifact of initial state averaging ("diffusion from convection"). This is also reflected in

the fact that the diffusive scale correction depends only on the two-point correlation function. To conclude, our work has clarified and justified recent developments in hydrodynamics of integrable models, such as BMFT and "diffusion from convection".

Since in hard rods we have not found any intrinsic diffusive correction, entropy is conserved for all times $t$. We have explained on the simple case of non-interacting classical particles that the hydrodynamic densities will eventually thermalize on a time scale dictated by coarse-graining (similar to [53]). This is in stark contrast to avoiding coarse-graining by averaging over the initial configuration: in this case, if hydrodynamics holds, the expectation value will follow the Euler hydrodynamic equation until diffusive time scales $t \sim \ell$, where the small diffusive effects become $\mathcal{O}(1)$. Interestingly, this means that hydrodynamics necessarily breaks down even before reaching diffusive time scales. Clarifying these different behaviors for $t \gg 1$ would be important to understand the relationship between and the limitations of the traditional approach to hydrodynamics and "hydrodynamics without averaging".

We would like to finish this paper by summarizing open scientific questions/problems that come out of this paper and that we believe will be useful to help to gain significantly deeper understanding into the hydrodynamics of both integrable and non-integrable models

- Developing a mathematical framework to study "hydrodynamics without averaging" so that one does not need to average over the initial state. For instance, it would be important to be able to quantify the precision of hydrodynamics using a proper norm. Also, can one make the requirement that configurations are locally "sufficiently generic" more precise. Note that such understanding could drastically simplify proving the emergence of hydrodynamics: one would only need to prove that the time-evolution maps "sufficiently generic" configurations onto "sufficiently generic" configurations and that hydrodynamics holds for a short time window $\Delta t$ with some rigorous error bounds. By iterating this argument $T/\Delta t$ times, one would immediately obtain that hydrodynamics applies for all times $T$.

- What happens if one applies "hydrodynamic without averaging" to non-integrable systems? Is Euler hydrodynamics still accurate on the diffusive scale or is there a true intrinsic diffusive correction? While, for now analytical treatments of non-integrable models are out-of-sight, we believe that already numerical simulations would give interesting insights.

- Does hydrodynamics apply to non-physical (i.e. not states with local equilibrium) also in other integrable or even in non-integrable models?

- To improve the accuracy of coarse-graining: Are there better coarse-grainings? Are there better choices of the densities? Also, what happens if one considers a non-smooth observable $\Upsilon(x, p)$? Is there any benefit from additionally coarse-graining in time?

- Due to the dependence on the coarse-graining scheme: In an experimental setup, which scheme should one use? Is there a natural scheme?

- Quantify the error of hydrodynamics: Can one predict the noise giving rise to the deviations of the microscopic dynamics from hydrodynamics? Is it Gaussian? How is it related to initial state noise? Also does this noise has an intrinsic component (in the sense of independent from coarse-graining and initial state noise)?

- It would be interesting to explore "hydrodynamics without averaging" in quantum systems. There are two different interpretations of a non-random state. One option is to consider pure states. While this does not have classical fluctuations, it still has quantum fluctuations, i.e. charge densities will not be deterministic. To circumvent this, one

could further restrict to pure states, that are locally eigenvectors of (coarse-grained) densities. These would then not fluctuate, neither classical nor quantum. Note, however that charge densities typically not commute, hence the best we can hope for are approximate (asymptotic) eigenstates of coarse-grained charge densities.

- Thermalization and entropy increase: is thermalization (from the hydrodynamic perspective) purely due to coarse-graining? Or is there a physical observer-independent process leading to it? For instance, we expect gradient catastrophes like shocks or turbulence to increase entropy. Apart from those, are there other processes (like diffusion) leading to entropy increase.

- Related to the above: It would be interesting to apply the ideas of "hydrodynamics without averaging" to purely diffusive systems (for instance the chaotic Ising chain [56]). Here, Euler hydrodynamics will be trivial (all currents vanish), hence there are no "diffusion from convection" effects. Also, there must be a physical process leading to intrinsic diffusion (otherwise the system would not be diffusive). Such systems would therefore be ideal to understand the emergence of diffusion in non-noisy many-body systems.

- Extending "hydrodynamics without averaging" to include the effect of large scale external potentials. In integrable models this is particularly interesting since an external potential breaks integrability and hence the system should thermalize. The Euler GHD equation with external potential however still conserves entropy [57, 58]. Therefore, is thermalization again only observed due to coarse-graining or is an actual physical effect (and on what time scale)? From this viewpoint, it would also be interesting to tackle the open problem of the numerically observed absence of thermalization of hard rods in an harmonic trap [40, 41].

## Acknowledgements

I would like to thank Benjamin Doyon for interesting discussions about these results and him and Leonardo Biagetti and Jacopo De Nardis for collaborations on related projects. The author acknowledges funding from the faculty of Natural, Mathematical & Engineering Sciences at King's College London. Numerical computations were done in Julia [59], and ran on the CREATE cluster [60].

## A   Details on the derivation of the diffusive GHD equation

Since the diffusive scale correction is linear in the two-point function, we can conveniently treat each component of the correlations separately.

### A.1   Singular part of the correlations

Let us start by considering the singular part of (131), i.e.

$$\langle \delta\rho(x,p)\delta\rho(y,q)\rangle = \tfrac{1}{\ell}\delta(x-y)C_{\text{GGE}}(x,p,q). \tag{A.1}$$

In this case we have the following identities

$$\ell \langle \delta\rho(x,p)\delta\rho(y,q)\rangle = \rho(x,p)\delta(x-y)C_{\text{GGE}}(x,p,q) \tag{A.2}$$

$$\langle \delta\rho(x,p)\delta\bar{\rho}(y)\rangle = \rho(x,p)\delta(x-y)1^{\text{dr}}(x)^2 \tag{A.3}$$

$$\ell \langle \delta\bar{\rho}(x)\delta\bar{\rho}(y)\rangle = 1^{\text{dr}^2}\bar{\rho}(x)\delta(x-y) \tag{A.4}$$

$$\ell \left\langle \delta\hat{X}(x)\delta\hat{X}(y)\right\rangle = a^2 \int_{-\infty}^{x\wedge y} \mathrm{d}z\, 1^{\text{dr}^2}(z)\bar{\rho}(z) =: a^2\Gamma(x\wedge y) \tag{A.5}$$

$$\ell \left\langle \delta\rho(y,q)\delta\hat{X}(x)\right\rangle = -a\rho(y,q)1^{\text{dr}}(y)^2\theta(x-y). \tag{A.6}$$

Here $x \wedge y = \min(x,y)$. We will need identity (A.6) evaluated at $x = y$ where $\theta(x-y)$ is singular. We will replace this by $1/2$, not only because it is natural, but also because it agrees with the coarse-graining procedure: since $\rho(x,p)$ is a coarse-grained quantity it is a "smeared out quantity" over the coarse-graining length scale $\Delta x$. Hence, $\int_{-\infty}^{x} \mathrm{d}y\, \rho(y,q)$ should be correlated exactly to the half of the fluid cell over which $\rho(x,p)$ is coarse-grained. Based on this we will use

$$\ell \left\langle \delta\rho(x,q)\delta\hat{X}(x)\right\rangle = -\tfrac{1}{2}a\rho(x,q)1^{\text{dr}}(x)^2. \tag{A.7}$$

Inserting these identities into (148), (154) and (155) we find

$$\Delta X(t,x,p) = -a^2 \int \mathrm{d}q\, \rho(y,q)1^{\text{dr}}(y)(\theta(x-y)-\tfrac{1}{2})\Big|_{y=X(\hat{X}(x)+(p-q)t)}$$

$$+ \tfrac{a^3}{2}\int \mathrm{d}q\, \frac{1}{1^{\text{dr}}(y)}\partial_y\Big(\frac{\rho(y,q)}{1^{\text{dr}}(y)}(\Gamma(x\vee y)-\Gamma(x\wedge y))\Big)\Big|_{y=X(\hat{X}(x)+(p-q)t)} \tag{A.8}$$

$$= \tfrac{a^2}{2}\int \mathrm{d}q\, \rho(y,q)1^{\text{dr}}(y)\,\text{sgn}(p-q)\Big|_{y=X(\hat{X}(x)+(p-q)t)}$$

$$+ \tfrac{a^3}{2}\int \mathrm{d}q\, \frac{1}{1^{\text{dr}}(y)}\partial_y\Big(\frac{\rho(y,q)}{1^{\text{dr}}(y)}(\Gamma(x\vee y)-\Gamma(x\wedge y))\Big)\Big|_{y=X(\hat{X}(x)+(p-q)t)} \tag{A.9}$$

and

$$\ell \langle \delta\rho(x,p)\delta X(t,x,p)\rangle = -\tfrac{1}{2}a\rho(x,p)1^{\text{dr}}(x)^2 + a\int \mathrm{d}q\, \theta(p-q)C_{\text{GGE}}(x,p,q)$$

$$- a^2\rho(x,p)1^{\text{dr}}(x)^2 \int \mathrm{d}q\, \frac{\rho(y,q)}{1^{\text{dr}}(y)}\big(\tfrac{1}{2}-\theta(y-x)\big)\Big|_{y=X(\hat{X}(x)+(p-q)t)} \tag{A.10}$$

$$= \tfrac{a}{2}\int \mathrm{d}q\, \text{sgn}(p-q)C_{\text{GGE}}(x,p,q) + \tfrac{a^2}{2}\rho(x,p)1^{\text{dr}}(x)^2 \int \mathrm{d}q\, \frac{\rho(y,q)}{1^{\text{dr}}(y)}\,\text{sgn}(p-q)\Big|_{y=X(\hat{X}(x)+(p-q)t)} \tag{A.11}$$

and

$$
\ell \left\langle \delta X_t(x,p)^2 \right\rangle = a^2 \Gamma(x)
$$

$$
+ a^2 \int dy\, dq\, dq'\, \theta(\hat{X}(x) + pt - \hat{X}(y) - qt)\theta(\hat{X}(x) + pt - \hat{X}(y) - q't)C_{\text{GGE}}(y,q,q')
$$

$$
+ a^4 \int dq\, dq'\, \frac{\rho(y,q)}{1^{\text{dr}}(y)} \frac{\rho(y',q')}{1^{\text{dr}}(y')}\big(\Gamma(x) - 2\Gamma(y) + \Gamma(y \wedge y')\big)\bigg|_{y=X(\hat{X}(x)+(p-q)t),y'=X(\hat{X}(x)+(p-q')t)}
$$

$$
- 2a^2 \int dy\, dq\, \rho(y,q)1^{\text{dr}}(y)^2\theta(\hat{X}(x) + pt - \hat{X}(y) - qt)\theta(x - y)
$$

$$
+ 2a \int dq\, \frac{\rho(y,q)}{1^{\text{dr}}(y)}(\Gamma(x) - \Gamma(y))\theta(x - y)\bigg|_{y=X(\hat{X}(x)+(p-q)t)}
$$

$$
- 2a^3 \int dy\, dq\, dq'\, \rho(y,q)1^{\text{dr}}(y)^2\theta(\hat{X}(x) + pt - \hat{X}(y) - qt)\frac{\rho(y',q')}{1^{\text{dr}}(y')}
$$

$$
\times \big(\theta(x - y) - \theta(y' - y)\big)\bigg|_{y'=X(\hat{X}(x)+(p-q')t)}. \tag{A.12}
$$

Now observe the following identities:

$$
\partial_y(\Gamma(x \vee y) - \Gamma(x \wedge y)) = \Gamma'(x \vee y)\theta(y - x) - \Gamma'(x \wedge y)\theta(x - y) \tag{A.13}
$$

$$
\partial_y^2(\Gamma(x \vee y) - \Gamma(x \wedge y)) = \Gamma''(x \vee y)\theta(y - x) + \Gamma'(x)\delta(x - y)
$$

$$
- \Gamma''(x \wedge y)\theta(x - y) + \Gamma'(x)\delta(x - y) \tag{A.14}
$$

$$
= \Gamma''(x \vee y)\theta(y - x) - \Gamma''(x \wedge y)\theta(x - y) + 2\Gamma'(x)\delta(x - y) \tag{A.15}
$$

and

$$
\partial_y(\Gamma(x \vee y) - \Gamma(x \wedge y))\bigg|_{y \to x} = \Gamma'(x)\,\text{sgn}(y - x) \tag{A.16}
$$

$$
\partial_y^2(\Gamma(x \vee y) - \Gamma(x \wedge y))\bigg|_{y \to x} = \Gamma''(x)\,\text{sgn}(y - x). \tag{A.17}
$$

Using them we can explicitly compute

$$
\lim_{t \to 0^+} \Delta X(t,x,p) = \frac{a^2}{2} \int dq\, \rho(x,q)\,\text{sgn}(p - q) \tag{A.18}
$$

and

$$
\lim_{t \to 0^+} \ell \left\langle \delta\rho(x,p)\delta X(t,x,p) \right\rangle = \frac{a}{2} \int dq\, \text{sgn}(p - q)\big[C_{\text{GGE}}(x,p,q) + a\rho(x,p)\rho(y,q)1^{\text{dr}}(x)\big] \tag{A.19}
$$

$$
= -\frac{a^2}{2}\rho(x,p) \int dq\, \rho(y,q)\,\text{sgn}(p - q) \tag{A.20}
$$

and

$$
\lim_{t \to 0^+} \ell \left\langle \delta X_t(x,p)^2 \right\rangle = a^2\Gamma(x) + a^2 \int dy\, dq\, dq'\, \theta(x - y)C_{\text{GGE}}(y,q,q')
$$

$$
- 2a^2 \int dy\, dq\, \rho(y,q)1^{\text{dr}}(y)^2\theta(x - y) = 0. \tag{A.21}
$$

From this we also get

$$\lim_{t \to 0^+} B(t,x,p) = 0. \tag{A.22}$$

Next, we can compute the time derivative

$$\frac{\mathrm{d}}{\mathrm{d}t}\Delta X(t,x,p)\bigg|_{t=0^+} = \frac{a^2}{2} \int \mathrm{d}q \, \partial_x(\rho(x,q)1^{\mathrm{dr}}(x))\frac{|p-q|}{1^{\mathrm{dr}}(x)}$$
$$+ \frac{a^3}{2}\int \mathrm{d}q \, \frac{\rho(y,q)}{1^{\mathrm{dr}}(y)^2}\Gamma''(x)\frac{|p-q|}{1^{\mathrm{dr}}(x)}$$
$$+ \frac{a^3}{2}\int \mathrm{d}q \left[\partial_x\left(\frac{\rho(x,q)}{1^{\mathrm{dr}}(x)^2}\right) + \frac{1}{1^{\mathrm{dr}}(x)}\partial_x\frac{\rho(x,q)}{1^{\mathrm{dr}}(x)}\right]\Gamma'(x)\frac{|p-q|}{1^{\mathrm{dr}}(x)}. \tag{A.23}$$

Defining $A(x,p) = \int \mathrm{d}q \, \rho(x,q)|p-q|$ this can be compactly written as

$$\frac{\mathrm{d}}{\mathrm{d}t}\Delta X(t,x,p)\bigg|_{t=0^+} = \frac{a^2}{2}\left[\frac{1}{1^{\mathrm{dr}}(x)}\partial_x A(x,p) + a\bar{\rho}(x)\partial_x\frac{A(x,p)}{1^{\mathrm{dr}}(x)}\right]. \tag{A.24}$$

Similarly, we find

$$\frac{\mathrm{d}}{\mathrm{d}t}\ell\left\langle\delta\rho(x,p)\delta X(t,x,p)\right\rangle\bigg|_{t=0^+} = \frac{a^2}{2}\rho(x,p)1^{\mathrm{dr}}(x)\partial_x\frac{A(x,p)}{1^{\mathrm{dr}}(x)} \tag{A.25}$$

and

$$\frac{\mathrm{d}}{\mathrm{d}t}\ell\left\langle\delta X_t(x,p)^2\right\rangle\bigg|_{t=0^+} = \frac{a^2}{1^{\mathrm{dr}}(x)}A(x,p). \tag{A.26}$$

Therefore,

$$\Delta j(x,p) = \frac{a^2}{2}\rho(x,p)\frac{\partial_x A(x,p)}{1^{\mathrm{dr}}(x)} - \frac{a^2}{21^{\mathrm{dr}}(x)}\partial_x\rho(x,p) = -\tfrac{1}{2}\hat{\mathbf{D}}\partial_x\rho, \tag{A.27}$$

which inserted into (144) gives (156).

## A.2   Long range part

Now, let us study long range correlation. We will assume that $C(x,p,y,q)$ is piecewise continuous but might have a jump at $x = y$. This in particular means that $\left\langle\delta\rho(y,q)\delta\hat{X}(x)\right\rangle$ is continuous and (weakly) differentiable. Let us write $C = C_{+1}(x,p,y,q)$ if $y > x$ and $C = C_{-1}(x,p,y,q)$ else. Then we have:

$$\ell\left\langle\delta\rho(y,q)(\delta\hat{X}(x) - \delta\hat{X}(y))\right\rangle = -a\int_y^x \mathrm{d}z \, \ell\left\langle\delta\rho(y,q)\bar{\rho}(z)\right\rangle \tag{A.28}$$

$$= -a\int_y^x \mathrm{d}z \int \mathrm{d}p \, C_{\mathrm{sgn}\,y-x}(z,p,y,q) \tag{A.29}$$

$$\ell\partial_y\left\langle\delta\rho(y,q)(\delta\hat{X}(x) - \delta\hat{X}(y))\right\rangle = a\int \mathrm{d}p \, C_{\mathrm{sgn}\,y-x}(y,p,y,q)$$

$$- a\int_y^x \mathrm{d}z \int \mathrm{d}p \, \partial_{x_2}C_{\mathrm{sgn}\,y-x}(z,p,y,q) \tag{A.30}$$

$$\ell\left\langle(\delta\hat{X}(x) - \delta\hat{X}(y))^2\right\rangle = a^2\int_y^x \mathrm{d}z_1\,\mathrm{d}q_1\,\mathrm{d}z_2\,\mathrm{d}q_2\,C_{\mathrm{sgn}\,z_2-z_1}(z_1,q_1,z_2,q_2). \tag{A.31}$$

It follows easily from the fact that $C(x, p, y, q)$ is a non-singular function that

$$\lim_{t \to 0^+} \Delta X(t, x, p) = 0 \tag{A.32}$$

$$\lim_{t \to 0^+} \ell \left\langle \delta \rho(x, p) \delta X_t(x, p) \right\rangle = 0 \tag{A.33}$$

$$\lim_{t \to 0^+} \ell \left\langle \delta X_t(x, p)^2 \right\rangle = 0. \tag{A.34}$$

Next, we need to compute the time derivatives, which are given by

$$\frac{\mathrm{d}}{\mathrm{d}t} \Delta X(t, x, p) \bigg|_{t=0^+} = a^2 \int \mathrm{d}q \, \mathrm{d}q' \, \tfrac{1}{1^{\mathrm{dr}}(x)^2} \Big( \delta(q - q') + a \tfrac{\rho(x,q)}{1^{\mathrm{dr}}(x)} \Big)$$
$$\times \int \mathrm{d}q'' \, C_{\mathrm{sgn}(p-q)}(x, q'', x, q')(p - q) \tag{A.35}$$

and

$$\frac{\mathrm{d}}{\mathrm{d}t} \ell \left\langle \delta \rho(x, p) \delta X(t, x, p) \right\rangle \bigg|_{t=0^+} = a \int \mathrm{d}q \, \mathrm{d}q' \, \tfrac{1}{1^{\mathrm{dr}}(x)} \Big( \delta(q - q') + a \tfrac{\rho(x,q)}{1^{\mathrm{dr}}(x)} \Big)$$
$$\times C_{\mathrm{sgn}(p-q)}(x, p, x, q')(p - q) \tag{A.36}$$

and

$$\frac{\mathrm{d}}{\mathrm{d}t} \ell \left\langle \delta X_t(x, p)^2 \right\rangle \bigg|_{t=0^+} = 0. \tag{A.37}$$

Combining these we find

$$\frac{\mathrm{d}}{\mathrm{d}t} B(t, x, p) \bigg|_{t=0^+} = \tfrac{a}{1^{\mathrm{dr}}(x)} \int \mathrm{d}q \, \mathrm{d}p' \, \mathrm{d}q' \, (p - q) \Big( \delta(p - p') + a \tfrac{\rho(x,p)}{1^{\mathrm{dr}}(x)} \Big)$$
$$\times \Big( \delta(q - q') + a \tfrac{\rho(x,q)}{1^{\mathrm{dr}}(x)} \Big) C_{\mathrm{sgn}(p-q)}(x, p', x, q'). \tag{A.38}$$

This gives rise to (157).

# B Further details on the numerical simulations

For the simulations we used hard rods diameter $a = 0.3$ and the observable

$$\Upsilon(x, p) = \eta(x, p) = \tfrac{1}{2\pi} e^{-\frac{1}{2}(x^2 + p^2)}, \tag{B.1}$$

which we also use for smoothing.

For each given $0 < \mu < 1$ and each $\ell$ we then average over $N_{\mathrm{Samples}}$ simulation, see Table 1. The $\ell$ were chosen as 11 uniformly spaced values in the interval given in Table 1 (including endpoints).

In the fluid cell case, for a given $\mu$ and $\ell$, the number of fluid cells in $x$ and $p$ is given by

$$N_{\mathrm{cell}} = \lfloor (10\mu)\ell^{1-\mu} \rfloor. \tag{B.2}$$

These cells are uniformly spaced between $x = [-5, 5]$ and $p = [-5, 5]$. For the smoothing we set $\Delta x = \Delta p$ to the size of these fluid cells.

The reason why it was necessary to choose different parameters is because the simulations were quite demanding (in particular diagonalizing large Ginibre matrices for the Ginibre

| Coarse-graining | Ensemble | $A$ | $\ell$ | $N_{\text{Samples}}$ |
|---|---|---|---|---|
| Fluid cell | LES/Poisson | 5 | $1000 - 10000$ | 10000 |
|  | Ginibre | 5 | $50 - 500$ | 1000 |
| Smoothing | LES/Poisson | 2 | $500 - 5000$ | 10000 |
|  | Ginibre | 1 | $200 - 2000$ | 1000 |

Table 1: Simulation parameters for the different cases

states). The microscopic evolution were always exactly simulated. To simplify the simulations of the hydrodynamic evolution, we did not exactly simulate it, but used approximations. To be precise for fluid cell coarse-graining we used (29), (49), (69) and (80) respectively to estimate the value of the hydrodynamic approximation. We know that these approximations have an error $\mathcal{O}(\Delta x^2)$ and thus not larger than the extracted scalings.

Computing observables after smoothing is even more demanding. Unless for the initial coarse-graining and the non-interacting time-evolution (where the smoothing can be seen as convolution on the observable), we do not know how to compute the obsevables after smoothing. For the contraction we therefore compute the difference formula (110) directly. This case is also the one that is the most demanding to simulate: we observe that it requires large system sizes to converge. This is why we reduced the amplitude $A$ in this case, meaning we simulate a less dense hard rods gas. For the contracting this turns out to be ok, but for the full time-evolution, where the computations would be even more involved, we would need to go to very small $A$. This would be a dilute hard rods gas, meaning that we would not probe the correct regime for hard rods (hydrodynamics requires finite density). Furthermore, for the full time-evolution we would need to find a way to compute (125) accurately and efficiently, which we need for (123). We do not know how to do this. This is why we decided against providing numerical data for the full evolution under smoothing.

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
