# Peer review of "Hydrodynamics without Averaging -- a Hard Rods Study"

_SciPost Physics_

## Round 2 · Referee Report · Anonymous (Referee 1) · 2025-10-10

Disclosure of Generative AI use

The referee discloses that the following generative AI tools have been used in the preparation of this report:

I was checking for the typos

Report

The article pose a very important question of the universality of the hydrodynamic approximation under different approximation schemes leading from a microscopic to an effective continuous evolution. The author introduces a novel concept of a "hydrodynamics without averaging" and rigorously estimates the errors caused by this approximation. The author finds no intrinsic diffusive correction at the 1/l scale. The author further explains that the 'diffusive behaviour' will be observed if one considers an average over an ensemble of initial states. While the results are new and interesting, many details remained unclear and the presentation could be greatly improved.

Requested changes

Main suggestions

1) In general, it would be good to add an additional section "main results", explaining and summarizing the results of each section

2) It will be also good to announce the results and explain the logic of derivation at the beginning of each section. And more importantly trying to exactly pronounce the starting point: equations and assumptions from which the rest is derived.

For instance:

2.1) I assume that the evolution of the coarse-grained system is further captured by Eq. 8, however it was never explicitly stated. It would be good to comment on that directly when writing down equations 76-79 and also 119-120.

Specific technical points

3) It will be good to discuss what are the author's expectations in case of a general integrable systems. Does the introduced approach allow to derive (or at least guess) the generalization of Eq. 159 to the case of a generic integrable system. If not, briefly comment why.

4) The definiton of z_{\alpha,\beta,\alpha^\prime, \beta^\prime} between 85 and 86 is inaccurate or deserves explanation, because the RHS of the definition does not contain \alpha^\prime, \beta^\prime, containing instead \alpha(x_j), \beta(x_j).

5) The estimation for the number of summation terms in Eq. 87 is not clear from the text, in particular what are the summation bounds. It is also not clear why z*{α,β,α′,β′} = O(∆x), naively it looks that for different cells α,α′ the difference \hat{x}_α-\hat{x}_α′ might be of order one, which is O(1).

6) The equations 159-160 deserve better explanation, in particular, it seems that they should be complemented by an equation for G_{LR,sym} to be complete. Without such a relation, it is not clear how to use 159-160 alone for the numerical simulation.

Minor formatting

7) as a minor suggestion: it will be good to use {multline} envitonment for a multiline equuations like 159-160 (and many others). Othervise both lines get enumerated, which is not common.

Recommendation

Ask for minor revision

---

## Round 2 · Referee Report · Anonymous (Referee 2) · 2025-10-11

Strengths

1) Creative approach with concrete examples 2) Detailed calculations 3) Introduces many open questions that the methods could be applicable to.

Weaknesses

1) The applicability of these methods to more complicated models might not be so straightforward.

Report

This paper carefully considers the hard rod model and demonstrates how the definition of a fluid cell can be understood and influenced both by the method used to coarse grain the field. Although many of the concepts have previously appeared in the literature a systematic study of exactly how the fluid cell emerges is novel and suggests new and exciting possibilities for future studies.

Requested changes

1) Minor typos (page 4 'jugding' instead of judging) 2) There is a missing Delta x in Eq. 26

3) In figure 3 the blank plot should be removed.

4) Under remark 9 the sentence: "If on the other hand, one is able to only measure observables averaged over many initial states, then one should use (160) instead." Is a bit unclear to me. I take it to mean that (160) captures the dynamics that emerge having averaged over the initial states. Perhaps this could be slightly reworded to clarify the difference between (8) (evolve many samples then average) vs (160) (average initial states then evolve).

5) The author points out Ginibre states as being particularly unphysical thus demonstrating the robustness of GHD. Does the author have any insight about the types of ensemble of states that are not captured by GHD?

Recommendation

Publish (easily meets expectations and criteria for this Journal; among top 50%)

---

## Round 2 · Referee Report · Anonymous (Referee 3) · 2025-10-13

**Report of the manuscript titled**

**"Hydrodynamics without Averaging - a Hard Rods Study'**

Instead of starting from a statistical state (ensemble), the article demonstrates how one can get hydrodynamic evolution in 1d hard-rod gas starting from a single microscopic configuration. The author does so by explicitly performing coarse graning of the particle phase space density and rigorously estimates the error in the 'hydrodynamic approximation'. The main finding of the paper is that there are not diffusive correction to the HD approximation and the Euler HD equation is valid even in the diffusive-space time scale. However, the there is a diffusive correlation when one starts from an ensemble of states and looks at the evolution of the average phase space density. The results are interesting and novel. I feel these results meet the criterion of the journal and I would like to recommend publication of the paper. However, I feel the presentation of manuscript can be improved and I request for a minor revision. Please find my comments below:

- (i) The presentation of the derivation in sections 3 and 4 can be improved. If possible, it would be useful to clearly mention the motivation, goal and the plan of the derivation at the beginning of each section+ subsection. Currently the conclusion of the sections are also not very clearly stated.

- (ii) In BMFT it is typically assumed that the for coarse-gaining scale $\ell_{cg} \sim \ell_{variation}$ and in such scale also the evolution of the initial fluctuations are described by Euler equations. For coarse graining size $\mu = 1/2 \ or' > 1/2$, it seems there will be significant noise. Does it imply one needs to look at such noise to understand (corrections to ) correlation at diffusive space-time scale?

- (iii) The proof of Eq. (137) is not given in Appendix A

- (iv) It would be good to write the representation of the operator $\hat{D}$ in Eq. (156).

- (v) Eq.(A1) and (A2) are same and (A2) has a typo.

- (vi) Eq. (A12): beginning of the second last line: $+2a \ \rightarrow \ 2a^3$

- (vii) Eq. (A27): $A(x,p)$ in the second term on the right.

- (viii) First line of sec. (A.2): the correlation should be $C_{LR}$ according to the notation in Eq. (131).

---

## Round 2 · Referee Report · Anonymous (Referee 4) · 2025-10-24

Strengths

  • Timely work on an important timely question: Is the assumption of local equilibrium really necessary to describe a system by hydrodynamics?

  • Careful study of a specific classical model: hard rods

  • Interesting statistical ensembles introduced in the numerical part: Poisson (no correlations) and Ginibre (correlated) distributions of hard rods in phase space

Weaknesses

  • The paper could probably be more concise. At times, it is hard to follow the author's reasoning.

  • While the message of the paper is formulated in a way that suggests that the idea applies to any system, regardless of its integrability, the chosen example (hard rods) is integrable, and the role of integrability in all of this remains quite unclear.

Report

This manuscript belongs to a series of recent works (see e.g. Refs. [16,17,18,54,55]) in the context of Generalized hydrodynamics that have asked the question: Is the assumption of local generalized equilibrium (local GGE) necessary for generalized hydrodynamics to hold on large distance and time scales ?

It provides a careful study of that question for the case of the classical hard rod gas. The conclusion is that no local generalized equilibrium assumption is needed.

I think the paper contains interesting ideas and it will motivate further works along this direction. I recommend publication in Scipost Physics after the author implements the changes below.

Requested changes

1- The introduction is very long, and not very focused. It could be more concise, and 'to-the-point'. A summary of the results would be appreciated. Also , in the first paragraph of the introduction the author seems to be talking about integrable systems since he is talking about generalized Gibbs ensembles. But then the later discussion in the introduction seems to be aimed at all systems, including non-integrable. Please clarify whether this introduction is about integrable or generic systems.

2- Sections 3.1 ,3.2,3.3,3.4 and 5.2 merely look like technical notes. Please provide a roadmap to the reader at the beginning of each subsection to explain what is the goal of the calculations

3- Please clarify whether / which conclusions of this paper hold beyond the integrable case.

4- 'Weak solution' in page 7 does not seem to be defined. Please define.

5-There are many typos ( p.2 'would be differ', p.4 'hards', p.8 'is that is that', 'corase-graining', etc). Please proofread.

Recommendation

Ask for minor revision

---

## Editorial Decision

awaiting_resubmission